# Influence of Bulk Microphysics Schemes upon Weather Research and Forecasting (WRF) Version 3.6.1 Nor'easter Simulations

Stephen D. Nicholls[1,2], Steven G. Decker[3], Wei-Kuo Tao[1], Stephen E. Lang[1,4], Jainn J. Shi[1,5], and Karen I. Mohr[1]

[1]*NASA-Goddard Space Flight Center, Greenbelt, 20716, United States of America*

[2]*Joint Center for Earth Systems Technology, Baltimore, NASA-Goddard Space Flight Center, Baltimore, 21250, United States of America*

3*Department of Environmental Sciences, Rutgers, The State University of New Jersey, 08850, United States of America*

[4]*Science Systems and Applications, Inc., Lanham, 20706, United States of America*

[5]*Goddard Earth Sciences Technology and Research, Morgan State University, 21251, United States of America*

*Correspondence to: Stephen D. Nicholls (stephen.d.nicholls@nasa.gov)*

**Abstract.** This study evaluated the impact of five, single- or double- moment bulk microphysics schemes (BMPSs) on Weather Research and Forecasting model (WRF) simulations of seven, intense winter-time cyclones impacting the Mid-Atlantic United States. Five-day long WRF simulations were initialized roughly 24 hours prior to the onset of coastal cyclogenesis off of the North Carolina coastline. In all, 35 model simulations (5 BMPSs and seven cases) were run and their associated microphysics-related storm properties (hydrometeor mixing ratios, precipitation, and radar reflectivity) were evaluated against model analysis and available gridded radar and ground-based precipitation products. Inter-BMPS comparisons of column-integrated mixing ratios and mixing ratio profiles reveal little variability in non-frozen hydrometeor species due to their shared programming heritage, yet their assumptions concerning snow and graupel intercepts, ice supersaturation, snow and graupel density maps, and terminal velocities lead to considerable variability in both simulated frozen hydrometeor species and radar reflectivity. WRF-simulated accumulated precipitation fields exhibit minor spatio-temporal variability amongst BMPSs, yet their spatial extent is largely conserved. Compared to ground-based precipitation data, WRF simulations demonstrate low-to-moderate (0.217–0.414) threat scores and a rainfall distribution shifted toward higher values. Finally, an analysis of WRF and gridded radar reflectivity data via contoured frequency with altitude (CFAD) diagrams reveals notable variability amongst BMPSs, where better performing schemes favored lower graupel mixing ratios and better underlying aggregation assumptions.

# 1 Introduction

Bulk microphysical parameterization schemes (BMPSs), within modern numerical weather prediction models (e.g., the Weather Research and Forecasting model [WRF; Skamarock et al., 2008]), have become increasingly complex and computationally expensive. Presently, the BMPS options offered in WRF vary from simplistic, warm rain physics (Kessler, 1969) to multi-phase, six-class, two-moment microphysics (Morrison et al., 2009). Microphysics and cumulus parameterizations drive cloud and precipitation processes within WRF and similar models, which has consequences for radiation, moisture, aerosols, and other simulated meteorological processes. Tao et al. (2011) highlighted the importance of BMPSs in models by summarizing more than 36 published, microphysics-focused studies ranging from idealized simulations to hurricanes to mid-latitude convection. More recently, the observation-based studies of Stark (2012) and Ganetis and Colle (2015) investigated microphysical species variability within United States (U.S.) East Coast winter-time cyclones (locally called "nor'easters") and have called for further investigation into how BMPSs impact these cyclones, which motivates this nor'easter study.

A "nor'easter" is a large (~2000 km), mid-latitude cyclone occurring from October to April and is capable of bringing punishing winds, copious precipitation, and potential coastal flooding to the Northeastern U.S. (Kocin and Uccellini 2004; Jacobs et al., 2005; Ashton et al., 2008). This region is home to over 65 million people and produces 16 billion U.S. dollars of daily economic output (Morath, 2016). Given its high economic output, nor'easter-related damages and disruptions can be extreme. Just ten strong December nor'easters between 1980 and 2011 produced 29.3 billion U.S. dollars in associated damages (Smith and Katz, 2013).

Recent nor'easter studies are scarce given the extensive research efforts of the 1980s. Those historical studies
addressed key environmental drivers including frontogenesis and baroclinicity (Bosart, 1981; Forbes et al., 1987;
Stauffer and Warner, 1987), anticyclones (Uccelini and Kocin, 1987), latent heat release (Uccelini et al., 1987), and
moisture transport by the low-level jet (Uccellini and Kocin, 1987; Mailhot and Chouinard, 1989). Despite extensive
observational analyses, little attention has been given to role of BMPSs in mid-latitude winter cyclones.
Reisner et al. (1998) ran several Mesoscale Model Version 5 winter storm simulations with multiple BMPS
options that impacted the Colorado Front Range during the Winter Icing and Storms Project. Double moment-based
simulations produced more accurate simulations of super cooled water and ice mixing ratios than those originating
from single-moment schemes. However, single moment-based simulations vastly improved when the snow-size
distribution intercepts were derived from a diagnostic equation rather than from a fixed value.
Wu and Pretty (2010) investigated how five, six-class BMPSs affected WRF simulations of four polar-low events
(two over Japan, two over the Nordic Sea). Their simulations yielded nearly identical storm tracks, but notable cloud
top temperature and precipitation errors. Overall, the WRF single-moment BMPS (Hong and Lim, 2006) produced
marginally better cloud and precipitation process simulations than those from other BMPSs. For warmer, tropical
cyclones, Tao et al. (2011) investigated how four, six-class BMPSs impacted WRF simulations of Hurricane Katrina.
Granted the steering currents were rather robust, but they found that BMPS choice minimally impacted storm track,
yet sea-level pressure (SLP) varied up to 50 hPa.
Shi et al. (2010) evaluated several WRF single-moment BMPSs for a lake-effect snow event. Simulated radar
reflectively and cloud top temperature validation revealed that WRF accurately simulated the onset, termination, cloud
cover, and band extent of a lake-effect snow event; however, snowfall totals at fixed points were less accurate due to
interpolation of the mesoscale grid. Inter-BMPS simulation differences were small because low temperatures and
weak vertical velocities prevented graupel generation. Reeves and Dawson (2013) investigated WRF sensitivity to
eight BMPSs for a December 2009 lake-effect snow event. Simulated precipitation rates and snowfall coverage were
particularly sensitive to the BMPSs because vertical velocities exceeded hydrometeor terminal fall speeds in half of
their simulations. Vertical velocity differences were attributed to differing frozen hydrometeor species assumptions
made by each BMPS such as snow density values, temperature-dependent snow-intercepts, and graupel generation
terms.
This study will evaluate WRF nor'easter simulations and their sensitivity to six- and seven-class BMPSs with a
focus on microphysical properties and precipitation. The remainder of this paper is divided into three sections. The
methodology and analysis methods are explained in section 2. The results are shown in section 3. Finally, the
conclusions, its implications, and prospects for future research are described in section 4.
**2 Methods**
**2.1 Study design**
WRF version 3.6.1 (hereafter W361) solves a set of fully-compressible, non-hydrostatic, Eulerian equations in
terrain-following coordinates (Skamarock et al., 2008). Figure 1 shows the four-domain WRF grid configuration for
this study using 45-, 15-, 5-, and 1.667-km horizontal grid spacing, respectively. Additionally, this configuration
includes 61 vertical levels, a 50-hPa (~20 km) model top, and two-way domain feedback; cumulus parameterization
is turned off for Domains 3 and 4, which are convection permitting. Notably, the location of Domain 4 adjusts for
each case (Fig. 1). Global Forecasting System model operational analysis (GMA) data was used for WRF boundary
conditions. The above model configuration (except for the 4th domain) and parameterizations are derived from
Nicholls and Decker (2015). Model parameterizations include:
▪ Longwave radiation: New Goddard Scheme (Chou and Suarez, 1999; Chou and Suarez, 2001)
▪ Shortwave radiation: New Goddard Scheme (Chou and Suarez, 1999)
▪ Surface layer: Eta similarity (Monin and Obukhov, 1954; Janjic, 2002)
▪ Land surface: NOAH (Chen and Dudhia, 2001)
▪ Boundary layer: Mellor-Yamada-Janjic (Mellor and Yamada 1982; Janjic 2002)
▪ Cumulus parameterization: Kain-Fritsch (Kain, 2004)
This study investigates the seven nor'easter cases described in Table 1 and shown in Fig. 1. These cases are
identical to those in Nicholls and Decker (2015) and represent a small, diverse sample of nor'easter events of varying
intensity and seasonal timing. In Table 1, the Northeast Snowfall Impact Scale (NESIS) value serves as proxy for
storm severity (1 = notable, 5 = extreme) and is based upon storm duration, population impacted, area affected, and
snowfall severity (Kocin and Uccellini, 2004). Early and late season storms (Cases 1, 2, and 7) did not have snow
and thus lack a NESIS rating.
Five-day, WRF simulations for this study were initialized 24 hours prior to the first precipitation impacts in the
highly populated Mid-Atlantic region and prior to the onset of rapid, coastal cyclogenesis off of the North Carolina
coastline. This starting point provides sufficient time to establish mesoscale circulations, surface baroclinic zones,
and sensible and latent heat fluxes (Bosart, 1981; Uccelini and Kocin, 1987; Kuo et al., 1991; Mote et al., 1997; Kocin
and Uccellini, 2004; Yao et al., 2008, Kleczek et al., 2014). The first nor'easter-associated precipitation impacts are
defined as the first hourly accumulation of 0.5 mm (~0.02 inch) registered from the New Jersey Weather and Climate
Network (D. A. Robinson, pre-print, 2005) related to the cyclone. A smaller threshold was not used to avoid capturing
isolated showers occurring well ahead of the primary precipitation shield.
To investigate BMPS influence upon W361 nor'easter simulations, five BMPS are used (Table 2). These BMPSs
include three six-class, three-ice, single-moment schemes (Lin [Lin6; Lin et al., 1983; Rutledge and Hobbs, 1984],
Goddard Cumulus Ensemble [GCE6; Tao et al., 1989; Lang et al., 2007], and WRF single moment [WSM6; Hong
and Lim 2006]), a seven-class, four-ice, single-moment Goddard Cumulus Ensemble scheme (GCE7; Lang et al. 2014;
Tao et al. 2016), and finally, the six-class, three-ice, WRF double-moment scheme (WDM6; Lim and Hong 2010)).
In total, 35 model simulations were completed (7 nor'easters times 5 BMPSs).
**2.2 Evaluation and analysis techniques**
Model evaluation efforts involved comparing WRF output against GMA, Stage IV precipitation (StIV; Fulton et
al. 1998; Y. Lin and K.E. Mitchell, preprints, 2005), and Multi-Radar/Multi-Sensor System (MRMS) 3D volume radar
reflectivity (Zhang et al. 2016). GMA offers six-hourly, gridded dynamical fields, including water vapor, with global
coverage. StIV is a six-hourly, 4-km resolution, gridded, combined radar and rain gauge precipitation product
covering the United States. Finally, MRMS has a two-minute, 1.3-km resolution, gridded 3D volume radar mosaic
product derived from S- and C-band radars covering the United States and Southern Canada (Zhang et al. 2016), which
is the operational successor to the National Mosaic and Multi-Sensor QPE (NMQ; Zhang et al. 2011) product. Both
StIV and MRMS, however, are limited by the detection range of their surface-based assets. All cross comparisons
between WRF and these evaluation data were conducted at an identical grid resolution.
Analysis of WRF microphysical, precipitation, and simulated radar output was comprised of three main parts:
precipitable mixing ratios and domain-averaged mixing ratio profiles, simulated precipitation, and simulated radar
reflectivity. Precipitable mixing ratios are calculated for six microphysical species (vapor, cloud ice, cloud water,
snow, rain, and graupel) using the equation for precipitable water:
$$PMR = \frac{1}{\rho g} \int_{P_{top}}^{P_{sfc}} w \, dp \qquad (1)$$

In Eq. (1), $PMR$ is the precipitable mixing ratio in mm; $\rho$ is the density of water (1,000 kg m$^{-3}$); $g$ is the
gravitational constant (9.8 m s$^{-2}$); $p_{sfc}$ is the surface pressure (Pa), $p_{top}$ is the model top pressure (Pa); $w$ is the mixing
ratio (kg kg$^{-1}$); $dp$ is the change in atmospheric pressure between model levels (Pa). Only water vapor PMR is
evaluated because all other hydrometeor species in GMA are nonexistent and ground- and space-based validation data
for each PMR hydrometeor species is lacking, especially over the data-poor North Atlantic (Li et al., 2008; Lebsock
and Su, 2014). Similarly, mixing ratio profiles will only be inter-compared amongst BMPSs because satellite-derived
cloud ice profile products (e.g., CloudSat 2C-ICE; Deng et al. 2013) do not directly overpass Domain 4 during coastal
cyclogenesis for any case. WRF-simulated precipitation fields and their distributions were evaluated against StIV;
simulation error was quantified via bias and threat score (critical success index; Wilks, 2011) values. Finally,
contoured frequency with altitude diagrams (CFADs; Yuter and Houze 1995) were used to validate WRF-simulated
radar reflectivity to MRMS similar to the radar validation efforts of Lang et al. (2011) and Lang et al. (2014). A
CFAD offers the advantage of preserving frequency distribution information, yet is insensitive to spatio-temporal
errors. Additionally, CFAD-based scores were calculated for each height level and with time using Eq (2).
$$CS = 1 - \frac{\sum |PDF_m - PDF_o|_h}{200} \qquad (2)$$

In (2), $CS$ is the CFAD score and $PDF_m$ and $PDF_o$ (%) are the probability density functions (PDF) at constant
height from WRF and MRMS, respectively. The CFAD score ranges between 0 (no PDF overlap) to 1 (identical
PDFs) (Lang et al., 2014).
**3. Results**
**3.1 Hydrometeor species analysis**
Figure 2 displays six classes (water vapor, cloud water, graupel, cloud ice, rain, and snow) of precipitable mixing
ratios (mm) from each WRF simulation and GMA, and Fig. 3 shows corresponding simulated radar reflectivity (no
MRMS on this date) at 4,000 m above mean sea level (AMSL) from Case 5, Domain 4 at 06 UTC February 2010. At
this time, storm track errors are negligible, the cyclone is centralized within Domain 4, and corresponding mixing
ratio profiles (Fig. 4) all show peak graupel mixing ratios around 4,000 m AMSL. Figure 5 shows the seven-case
composite mixing ratio profiles derived from hourly data during the residence time for each nor'easter case within
Domain 4 (24-30 hours). This composite illustrates that mixing ratio profiles largely preserve their shape, maximum
mixing ratio heights, and mixing ratio tendencies (i.e., higher snow mixing ratios in GCE6 and GCE7), but hourly
mixing ratio values themselves can vary up to 3.5 times higher (e.g., QRAIN in WDM6) at a given height than in the
seven case composite (Fig. 5). Figures 4 and 5 also contain two black dashed lines denoting the 0°C and -40°C heights,
which denote the region where super-cooled water may occur. Although both the super-cooled water fraction and
these temperature heights vary hourly, the latter demonstrates little to no inter-BMPS variability. As seen in Fig. 4,
all cloud water and rain between 3,500 m and 10,000 m AMSL is super-cooled. Stronger nor'easter-related convection
(reflectivity > 35 dBZ) in Fig. 3 best corresponds to precipitable rain and then graupel (Fig. 2) despite the near non-
existence of the former at 4,000 m AMSL (Fig. 4). This apparent discrepancy is indicative of shallow convection
where liquid precipitable mixing ratios from the surface up to near the freezing level can well exceed those of frozen
hydrometeor species (i.e., graupel does not extend over a deep layer except within the convective line). Within the
broader precipitation shield (20-35 dBZ), radar reflectivity patterns best correspond to precipitable snow and then
precipitable graupel (Fig. 2) for all BMPSs except for Lin6 where this trend is reversed. Although Fig. 4 shows that
all five BMPSs loosely agree on the amount and height of maximum graupel content at 4,000 m AMSL, Lin6 has little
to any snow at this level, which likely explains the trend reversal. Inter-BMPS mixing ratio variability both at this
level and throughout the troposphere is associated with differing underlying assumptions made by each BMPS and is
explained in more detail below.

All evaluated BMPSs share a common heritage with the Lin scheme (Note: Lin6 is a modified form of the original

Lin scheme). Amongst the BMPSs, only WDM6 explicitly forecasts cloud condensation nuclei, rain, and cloud water
number concentrations, the remaining schemes apply derivative equations for these quantities (Hong et al., 2010).
Aside from the above, all five BMPS differ primarily in their treatment of frozen hydrometeors, which is most evident
from the nearly identical (exception: WDM6) rain mixing ratio profiles (Figs. 4 and 5) and precipitable water vapor
(Fig. 2) and is a result consistent with Wu and Petty (2010). Comparing WSM6 to WDM6 reveals the second moment
has little to no effect on precipitable rain coverage area (Fig. 2), yet WDM6 rainfall mixing ratios below the freezing
level are higher than in WSM6, except near the surface (Figs. 4 and 5). Min et al. (2015) ran WRF simulations of post-
monsoonal convection using WSM6 and WDM6 and generated similar rainfall mixing ratio profiles. They attribute
the profile differences to the capability of WDM6 to simulate the sedimentation processes of raindrops due to its
inclusion of the second moment and cloud condensation nuclei.

Similar to rain, precipitable cloud water extent (Fig. 2) and maximum cloud water height (Figs. 4 and 5) barely

change, yet mixing ratio amounts (Figs. 2, 4, 5) did vary amongst the BMPSs. These cloud water mixing ratio
differences are likely associated with both varying ice supersaturation allowances as described for the Goddard
schemes by Chern et al. (2016) and for the WRF schemes by Hong et al. (2010) and assumed cloud water number
concentrations (300 cm$^{-3}$ for WSM6). Although WDM6 borrows much of its source code from WSM6, forecasts of
cloud condensation nuclei and cloud water number concentrations alter inter-hydrometeor species interactions, which
in turn alter cloud water mixing ratios (Hong et al. 2010).  The similarly between WSM6 and WDM6 in Figs. 2-4
indicate that forecasted cloud number concentrations for Case 5 are likely close to the 300 cm$^{-3}$ value assumed by
WSM6.  For the other cases, cloud water mixing ratios did vary between WSM6 and WDM6 indicating that WDM6
cloud water number concentrations did likely stray from 300 cm$^{-3}$ and therefore cause the apparent differences in
composite cloud water mixing ratios (Fig. 5).

Figures 2, 4, and 5 show that precipitable snow and snow mixing ratios vary considerably amongst the BMPSs

with Lin6 and GCE6 having the smallest and largest snow amounts, respectively.  Dudhia et al. (2008) and Tao et al.
(2011) attribute the low snow mixing ratios in Lin6 to its high rates of dry collection of snow by graupel, its low snow
size distribution intercept (decreased surface area), and its auto-conversion of snow to either graupel or hail at high
mixing ratios.  In GCE6, the dry collection of snow and ice by graupel is turned off, greatly increasing the snow
mixing ratios at the expense of graupel, while the snow riming efficiency is reduced relative to Lin6 (Lang et al. 2007).
Snow growth in GCE6 is further augmented by its assumption of water saturation for the vapor growth of cloud ice
to snow (Reeves and Dawson, 2013; Lang et al. 2014).  In GCE7, the vapor growth issue in GCE6 is addressed with
a relative humidity (RH)-based correction factor; a snow size and density mapping, snow breakup interactions, and a
new vertical-velocity-dependent ice super saturation assumption are also introduced (Lang et al., 2007; Lang et al.,
2011; Lang et al., 2014; Chern et al., 2016; Tao et al., 2016).  Despite the reduced efficiency of vapor growth of cloud
ice to snow due to both the new RH correction factor and the ice super saturation adjustment, the new snow mapping
and enhanced cloud ice-to-snow auto-conversion in GCE7 offset this potential reduction, which kept GCE snow
mixing ratios higher than those in non-GCE BMPSs.  Unlike Lin6, in WSM6 and WDM6, grid cell graupel and snow
fall speeds are assumed to be identical (Dudhia et al., 2008) and that ice nuclei concentration is a function of
temperature (Hong et al., 2008).  These two aspects effectively eliminate the accretion of snow by graupel and increase
snow mixing ratios at lower temperatures (Dudhia et al., 2008; Hong et al., 2008).  Figures 4 and 5 show the maximum
snow mixing ratio height is roughly conserved in all non-Lin6 BMPSs.  Non-uniform graupel and snow fall speeds
and dry collection of snow by graupel in Lin6 reduces its snow mixing ratios in the middle troposphere and raises its
maximum snow mixing ratio height.

Compared to snow, graupel mixing ratios are generally smaller except for Lin6 where the dry collection of snow

by graupel leads to an unrealistic graupel-dominated scenario (Stith et al. 2002).  Graupel mixing ratios are lowest in
GCE7 due to the net effect of its additions (compared to GCE6) despite the inclusion of a new graupel size map.  In
particular, the combination of the new snow size mapping (decreased snow sizes aloft, increases snow surface area,
and enhances vapor growth), the addition of deposition conversion processes (graupel/hail particles experiencing
deposition growth at lower temperatures are converted to snow), and a reduction in super cooled droplets available
for riming (cloud ice generation is augmented, see below) all favor snow growth at the expense of graupel (Lang et
al. 2014; Chern et al., 2016; Tao et al., 2016).  Consistent with Reeves and Dawson (2013), WSM6 and WDM6 graupel
mixing ratio values are typically 30-50 % of their snow counterparts.

Although cloud ice mixing ratios are nearly an order of magnitude smaller than those for snow (GCE6), these

mixing ratios still vary greatly amongst the BMPSs as illustrated in Figs. 2, 4, and 5.  Cloud ice mixing ratios are
highest in GCE7 and lowest in Lin6.  Wu and Petty (2010) similarly found low cloud ice mixing ratios in Lin6
simulations and ascribe it to dry collection of cloud ice by graupel and its fixed cloud-ice size distribution. Similar to
Lin6, a monodispersed cloud-ice size distribution (20 μm diameter) is used in GCE6; however, in the vapor growth
of cloud ice to snow, water saturation conditions are still assumed even though ice supersaturation is not permitted.
As a result, excess vapor is first forced to cloud ice via the saturation adjustment scheme before being excessively
converted to snow (Lang et al., 2011; Tao et al., 2016) due to the assumption of water saturation in the growth of
cloud ice to snow term. In GCE7, the cloud ice-to-snow conversion rates are constrained using a RH-correction factor,
which is dependent upon ice supersaturation, which is itself dependent up vertical velocity. Additionally, GCE7 also
includes contact and immersion freezing terms (Lang et al., 2011), makes the cloud ice collection by snow efficiency
a function of snow size (Lang et al., 2011; Lang et al., 2014), sets a maximum limit on cloud-ice particle size (Tao et
al., 2016), makes ice nuclei concentrations follow the Cooper curve (Cooper, 1986; Tao et al., 2016), and allows cloud
ice to persist in ice subsaturated conditions (i.e., where RH for ice $\geq$ 70%) (Lang et al., 2011; Lang et al., 2014).
Despite the increased cloud ice-to-snow auto conversion rates in GCE7 (Lang et al. 2014; Tao et al. 2016), precipitable
cloud ice amounts nearly doubled relative to GCE6 (See Fig. 2). Similar to GCE7, larger cloud ice mixing ratios are
generated in WSM6 than in Lin6, which Wu and Petty (2010) attribute to excess cloud glaciation at temperatures
between 0°C and -20°C and its usage of fixed cloud ice size intercepts. Additionally, both WSM6 and WDM6 include
ice sedimentation terms, which promote smaller cloud ice amounts (Hong et al., 2008). Despite their varying
assumptions, the maximum cloud ice heights for both Case 5 and overall (Figs. 4 and 5) are consistent between the
five BMPSs.
**3.2 Stage IV precipitation analysis**
Excessive precipitation, whether frozen or not, is one of the most potentially crippling impacts of a nor'easter.
Figures 6 and 7 show Domain 3, accumulated precipitation, their difference from StIV, and the associated probability
and cumulative distribution functions (PDF and CDF, respectively) for Cases 5 and 7 based upon the 24-30 hour
residence period of a nor'easter within Domain 4. Domain 3 serves are the focus for this section because most of
Domain 4 resides close to or outside the StIV data boundaries. Cases 5 and 7 are chosen because of their near-shore
tracks (Fig. 1), which affords good StIV data coverage. Table 3 includes threat score and bias information from all
seven cases and their associated standard deviation statistics. Both threat score and model bias assume the same 10
mm threshold value, which is approximately the 25[th] percentile of accumulated precipitation (Figs. 6 and 7).
Case 4 threat score and bias values (Table 3) are more than two standard deviations from the composite mean due
to its non-coastal storm track (Fig. 1), and thus it is excluded from this analysis. The remaining six cases show WRF
to have low-to-moderate forecast skill (Threat scores: 0.217 [Lin6] – 0.414 [Lin6]) and to cover too large of an area
with precipitation accumulations greater than 10 mm (bias: 1.47 [Lin6, Case 7] – 4.05 [GCE7, Case 3] times the
observed area) relative to StIV. Inter-BMPS threat scores and bias differences are an order of magnitude or less than
the values from which they are derived. Consistent with Hong et al. (2010), threat score and bias values from WSM6
are equal to or improved upon by WDM6 due to its inclusion of a cloud condensation nuclei feedback. Overall, WDM6
shows marginally better precipitation forecast skill than the other BMPSs (highest threat score in four out of six cases
and highest mean threat score: 0.322), yet Lin6 is the least biased (lowest bias score in four of out of six cases and
lowest mean bias: 2.55).
PDF and CDF plots from Figs. 6 and 7 show WRF to favor higher precipitation amounts and is consistent with
the positive bias scores in Table 3. Previous modeling studies of strong convection by Ridout et al. (2005) and
Dravitzki and McGregor (2011) found that both GFS and the Coupled Ocean/Atmosphere Mesoscale Prediction
System produced too much light precipitation and too much heavy precipitation, which contrast with the above results.
Unlike these two studies, nor'easters track too far offshore to be fully sampled by rain gauge data and S-band weather
radars. These two issues could lead to an under bias in StIV data, especially near the data boundaries and suggests
that WRF threat scores and biases are likely closer to observations than as indicated in Table 3. Marginal changes in
accumulated precipitation PDFs and CDFs and threat scores amongst BMPSs are consistent with the investigation of
simulated precipitation during warm-season precipitation events and a quasi-stationary front by Fritsch and Carbone
(2004) and Wang and Clark (2010), respectively.
**3.3 MRMS and radar reflectivity analysis**
Figure 8 shows Domain 3, Case 4 radar reflectivity CFADs constructed during the 24-hour residence time of the
nor'easter within Domain 4 (12 UTC 26–27 January 2015). Domain 4 CFADs are not shown here because NOAA
radar quality control measures for non-precipitating echoes tend to artificially curtail radar echoes at 5 dBZ, especially
near the dataset edges (Jian Zhang, NOAA, personal communication). Domain 4-based CFADs (not shown) depict
little to no aggregation and are inconsistent with CFADs from previous convection (Lang et al. 2011, Min et al. 2015)
and mid-latitude winter storm (Shi et al. 2010) studies. The larger spatial extent and better radar overlap in Domain
3 leads to more realistic CFADs with aggregation. Case 4 data are shown in Fig. 8 because MRMS data were more
readily available and also based on the latest MRMS reprocessing algorithm.
Figure 8 shows that the MRMS-based CFAD has two distinct frequency maxima: one above and another below
6,000 m AMSL. Model simulations can replicate the sub-6,000 m AMSL frequency maxima with varying degrees of
success. Below 2,000 m (0°C height), GCE7- and Lin6-based CFADs more closely match the MRMS radar
reflectivity probability spectra and correctly show its maximum to occur between 0 and 15 dBZ. Other schemes over
broaden this probability spectra and shift its maximum toward higher reflectivity values. Despite this rightward shift,
hydrometeor profiles below 2,000 m AMSL (Fig. 4) are similar for all five of the BMPSs, which suggests that factors
including the assumed or simulated (WDM6) droplet size distributions or aggregation assumptions may be probable
causes.
Between 2,000 and 6,000 m AMSL all non-GCE7 CFADs incorrectly shift toward higher reflectivity values with
increasing height and favor values up to 10 dBZ higher (WSM6) than MRMS. Radar reflectivities at 3,000 m AMSL
on 26 January 2015 (Fig. 9) indeed show an overestimation of radar reflectivities in non-GCE7 BMPSs from regions
of strong convection off of the North Carolina and New Jersey coastlines near the cold front and warm front,
respectively. This rightward bowing of CFADs above the melting layer was also reproduced in Shi et al. (2010)
(GCE6) and Min et al. (2015) (WSM6 and WDM6). Similar to these studies, all non-GCE7 schemes seemingly
produce too much graupel (Figs. 4 and 5), which results in stronger reflectivity signatures (See section 3.1). GCE7
has the least graupel as a consequence of its new snow size mapping, inclusion of deposition-growth conversion
processes, reduced super-cooled cloud droplets and cloud-ice size restrictions.

Above 6,000 m AMSL the WRF-based CFADs all collapse toward smaller reflectivity values. This collapse is
well documented in the literature (Shi et al. 2010; Lang et al. 2011; Min et al. 2015) and occurs at least partly due to
errors stemming from increased entrainment of ambient air near cloud top and the underlying aggregation assumptions
made by each BMPS. Although each scheme fully collapses by 7,500 m AMSL, the Goddard-based CFADs indicate
a considerably steeper tilt in the maximum frequency core as compared to other schemes, which is a likely byproduct
of their higher snowfall mixing ratios (Fig. 4). Above 8,000 m AMSL, MRMS radar reflectivity values show a second
frequency maximum above 15 dBZ, which is not replicated by WRF. Radar reflectivities at 9,000 m AMSL on 26
January 2015 (Fig. 10) show precipitating echoes to occur offshore where the non-precipitating echo filtering applied
in MRMS removed weak reflectivities, artificially shifting the CFAD toward higher values.

Finally, CFAD scores (Eq. 2) with height and time (Fig. 11) provide a means to evaluate hourly forecast skill at
upper levels relative to MRMS. Figure 11 shows Lin6 and GCE7 to have notably improved forecast skill, especially
between 2,000 (0°C height) and 4,850 m AMSL compared to the other BMPSs. Despite their similar CFAD scores,
a properly oriented aggregation structure in its CFAD (Fig. 8) and better overall 3,000 m AMSL radar reflectivity
values (Fig. 9) suggests that GCE7 produces more realistic results than Lin6, which has unrealistically high graupel
growth rates due to the dry collection of snow. In short, Lin6 produces the right answer for the wrong reason, whereas
GCE7 produces the correct answer from a more realistic solution. Between 6,300 and 7,000 and m AMSL, GCE7
CFAD scores fall below the other schemes as a consequence of overly small particles from its size mapping and cloud
entrainment, associated with generally lower cloud tops. The other six cases produce similar tendencies in their CFAD
and CFAD scores as noted above for Case 4, except that cloud heights reach higher altitudes and CFADs become
wider with the introduction of stronger convection in early and late season events.
**4 Conclusions**

The role and impact of five bulk microphysics schemes (BMPSs; Table 2) upon seven Weather Research and
Forecasting model (WRF) winter time cyclone ("nor'easter") simulations (Table 1) are investigated and validated
against GFS model analysis (GMA), Stage IV rain gauge and radar estimated precipitation, and the radar-derived,
Multi-Radar/Multi-Sensor System (MRMS) 3D volume radar reflectivity product. Tested BMPSs include three
single-moment, six class BMPSs (Lin6, GCE6, and WSM6), one single-moment, seven class BMPS (GCE7), and one
double-moment, six-class BMPS (WDM6). Simulated hydrometer mixing ratios from single-moment BMPSs show
general similarities for non-frozen hydrometeor species (cloud water and rain) due to their common Lin6 heritage.
The inclusion of a double moment and cloud condensation nuclei permitted WDM6 to simulate the sedimentation
processes of raindrops, which increased rain mixing ratios below the freezing level relative to single-moment BMPSs.
Frozen hydrometeor species (snow, graupel, cloud ice) demonstrate considerably larger variability amongst BMPSs.
This variability results from differing assumptions concerning snow and graupel intercepts, degree of allowable ice
supersaturation, snow and graupel density maps, and terminal velocities made in each BMPS. WRF-simulated
precipitation fields exhibit similar coverage but trended towards higher precipitation amounts relative to Stage IV
observations resulting in low-to-moderate threat scores (0.217–0.414). Inter-model differences were an order of
magnitude or less than the threat score values, but WDM6 did demonstrate marginally better precipitation forecast
skill overall. Finally, MRMS-based contoured frequency with altitude diagrams (CFADs) and CFAD scores show
Lin6 and GCE7 to perform the best in the lower half of the troposphere (below 6,300 m AMSL), where GCE7 most
realistically reproduced the maximum frequency core between 5 and 15 dBZ due to its temperature and mixing ratio
dependent aggregation and new snow size mapping. However, the overly large growth of graupel via its dry collection
of snow suggests that Lin6 obtains high CFAD scores from a less realistic solution than GCE7. Above 6,300 m
AMSL, model-simulated cloud tops are much more susceptible to entrainment and become more sporadic; this in
conjunction with the non-precipitating echo filtering in the MRMS data makes evaluations less meaningful with
increasing height.
This study has shown that although cloud microphysics lead to only subtle differences in the large-scale
environment, they do noticeably alter the microphysical and precipitation properties of a nor'easter. While no BMPS
has consistently improved precipitation forecast skill, their underlying assumptions result in varying forecast skill of
simulated radar reflectivity structures between individual BMPSs when compared to MRMS observations. Follow-
on studies should investigate additional nor'easter cases or compare these cyclones to other weather phenomena (polar
lows, monsoon rainfall, drizzle, etc.). Results covering multiple phenomena may provide guidance to model users in
their selection of BMPS for a given computational cost. Additionally, potential studies could focus on the key aspects
of a nor'easter's structure (such as the low-level jet) or validation of model output against current and recently
available satellite-based datasets from MODIS (Justice et al., 2008), CloudSat (Stephens et al., 2008), CERES, and
GPM (Hou et al. 2014). Finally, other validation methods including object-oriented (Marzban and Sandgathe, 2006)
or fuzzy verification (Ebert 2008) could be utilized.
**5 Code availability**
WRF version 3.6.1 is publically available for download from the WRF Users' Page (http://www2.mmm.ucar.edu/
wrf/users/download/get_sources.html).
**6 Data availability**
GFS model analysis data boundary condition data can be obtained from NASA's open access NOMADS data
server (ftp://nomads.ncdc.noaa.gov/GFS/Grid3/). Stage IV precipitation data is publically available from the National
Data and Software Facility at the University Center for Atmospheric Research (http://data.eol.ucar.edu/cgi-
bin/codiac/fgr_form/id=21.093). Daily MRMS data is available from the National Severe Storms Laboratory
(http://www.nssl.noaa.gov/projects/mrms/).
**7 Author contributions**
S. D. Nicholls designed and ran all model simulations and prepared this manuscript. S. G. Decker supervised S.
D. Nicholls' research efforts, funded the research, and revised the manuscript. W.-K. Tao, S. E. Lang, and J. J. Shi
brought their extensive knowledge and expertise on model microphysics, which helped shape the project methodology
and rationalize the results. S. E. Lang also aided S. D. Nicholls in revising the manuscript and reviewer responses.
Finally, K. I. Mohr helped to facilitate connections between the research team and supervised S. Nicholls' research.
**8 Acknowledgements**
This research was supported by the Joint Center for Earth Systems Technology (JCET), the University of
Maryland Baltimore County (UMBC), and in part by the New Jersey Agricultural Experiment Station. Resources
supporting this work were provided by the NASA High-End Computing (HEC) Program through the NASA Center
for Climate Simulation (NCCS) at Goddard Space Flight Center.

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

**Table 1.** Nor'easter case list. The NESIS number is included for storm severity reference. Mean sea-level pressure
(MSLP) indicates maximum cyclone intensity in GMA. The last two columns denote the first and last times for each
model run. GMA storm tracks are displayed in Fig. 1.

| Case Number | NESIS | MSLP (hPa) | Event Dates | Model Run Start Date | Model Run End Date |
|---|---|---|---|---|---|
| 1 | N/A | 991.5 | 15–16 Oct 2009 | 10/15 00UTC | 10/20 00UTC |
| 2 | N/A | 989.5 | 07–09 Nov 2012 | 11/06 18UTC | 11/11 18UTC |
| 3 | 4.03 | 972.6 | 19–20 Dec 2009 | 12/18 18UTC | 12/23 18UTC |
| 4 | 2.62 | 980.5 | 26–28 Jan 2015 | 01/25 12UTC | 01/30 12 UTC |
| 5 | 4.38 | 979.7 | 05–07 Feb 2010 | 02/05 06UTC | 02/10 06UTC |
| 6 | 1.65 | 1005.5 | 02–03 Mar 2009 | 03/01 00UTC | 03/06 00UTC |
| 7 | N/A | 993.5 | 12–14 Mar 2010 | 03/11 18UTC | 03/16 18UTC |


**Table 2.** Applied bulk microphysics schemes and their characteristics. The below table indicates simulated mixing
ratio species and number of moments. Mixing ratio species include: QV = water vapor, QC = cloud water, QH = hail,
QI = cloud ice, QG = graupel, QR = rain, QS = snow.

| Microphysics Scheme | QV | QC | QH | QI | QG | QR | QS | Moments | Citation |
|---|---|---|---|---|---|---|---|---|---|
| Lin6 | X | X | | X | X | X | X | 1 | Lin et al. (1983); Rutledge and Hobbs (1984) |
| GCE6 | X | X | | X | X | X | X | 1 | Tao et al. (1989); Lang et al. (2007) |
| GCE7 | X | X | X | X | X | X | X | 1 | Lang et al. (2014) |
| WSM6 | X | X | | X | X | X | X | 1 | Hong and Lim (2006) |
| WDM6 | X | X | | X | X | X | X | 2 (QC, QR) | Lim and Hong (2010) |


**Table 3**. Stage IV-relative, accumulated precipitation threat scores and biases assuming a threshold value of 10 mm
(25[th] percentile of 24 hour accumulated precipitation). Bolded value denote the model simulation with the threat score
closest to 1 (perfect forecast) or a bias values closest to 1 (number of forecasted cells matches observations). The
lower two panels indicate the number of standards deviations (stdev) each threat score and bias value deviates from
the composite (all models + all cases) mean.

Domain 3

| Threat Score | 1 | 2 | 3 | 4 | 5 | 6 | 7 | Mean | Mean w/o 4 |
|---|---|---|---|---|---|---|---|---|---|
| Lin6 | 0.289 | 0.217 | 0.291 | 0.091 | **0.414** | **0.304** | 0.332 | 0.277 | 0.308 |
| GCE6 | 0.286 | **0.243** | 0.320 | 0.091 | 0.406 | 0.291 | 0.356 | 0.285 | 0.317 |
| GCE7 | 0.288 | 0.235 | 0.319 | **0.096** | 0.405 | 0.300 | 0.337 | 0.283 | 0.314 |
| WSM6 | **0.293** | 0.237 | 0.315 | 0.093 | 0.404 | 0.292 | 0.356 | 0.284 | 0.316 |
| WDM6 | 0.290 | **0.243** | **0.329** | 0.094 | 0.411 | 0.299 | **0.357** | **0.289** | **0.322** |

| Bias | 1 | 2 | 3 | 4 | 5 | 6 | 7 | Mean | Mean w/o 4 |
|---|---|---|---|---|---|---|---|---|---|
| Lin6 | 2.47 | **3.53** | **2.72** | 7.82 | **2.22** | 2.9 | **1.47** | 3.30 | 2.55 |
| GCE6 | **2.37** | 3.88 | 2.85 | 8.09 | 2.26 | 2.93 | 1.64 | 3.43 | 2.66 |
| GCE7 | 2.52 | 4.05 | 2.85 | **7.75** | 2.23 | 2.82 | 1.57 | 3.34 | 2.67 |
| WSM6 | 2.47 | 3.75 | 2.86 | 8.13 | 2.26 | 2.93 | 1.62 | 3.43 | 2.65 |
| WDM6 | **2.37** | 3.8 | 2.76 | 8.09 | 2.23 | **2.82** | 1.57 | **3.38** | **2.59** |

| T. Score Stats: | All Stdev | 0.094 | All Mean | 0.284 | | | |
|---|---|---|---|---|---|---|---|
| Threat Score | 1 | 2 | 3 | 4 | 5 | 6 | 7 |
| Lin6 | 0.06 | -0.71 | 0.08 | -2.05 | 1.39 | 0.22 | 0.52 |
| GCE6 | 0.03 | -0.43 | 0.39 | -2.05 | 1.31 | 0.08 | 0.77 |
| GCE7 | 0.05 | -0.52 | 0.38 | -2.00 | 1.29 | 0.18 | 0.57 |
| WSM6 | 0.10 | -0.50 | 0.34 | -2.03 | 1.28 | 0.09 | 0.77 |
| WDM6 | 0.07 | -0.43 | 0.48 | -2.02 | 1.36 | 0.16 | 0.78 |

| Bias Stats | All Stdev | 2.007 | All Mean | 3.389 | | | |
|---|---|---|---|---|---|---|---|
| Bias | 1 | 2 | 3 | 4 | 5 | 6 | 7 |
| Lin6 | -0.46 | 0.07 | -0.33 | 2.21 | -0.58 | -0.24 | -0.96 |
| GCE6 | -0.51 | 0.24 | -0.27 | 2.34 | -0.56 | -0.23 | -0.87 |
| GCE7 | -0.43 | 0.33 | -0.27 | 2.17 | -0.58 | -0.28 | -0.91 |
| WSM6 | -0.46 | 0.18 | -0.26 | 2.36 | -0.56 | -0.23 | -0.88 |
| WDM6 | -0.51 | 0.21 | -0.31 | 2.34 | -0.58 | -0.28 | -0.91 |


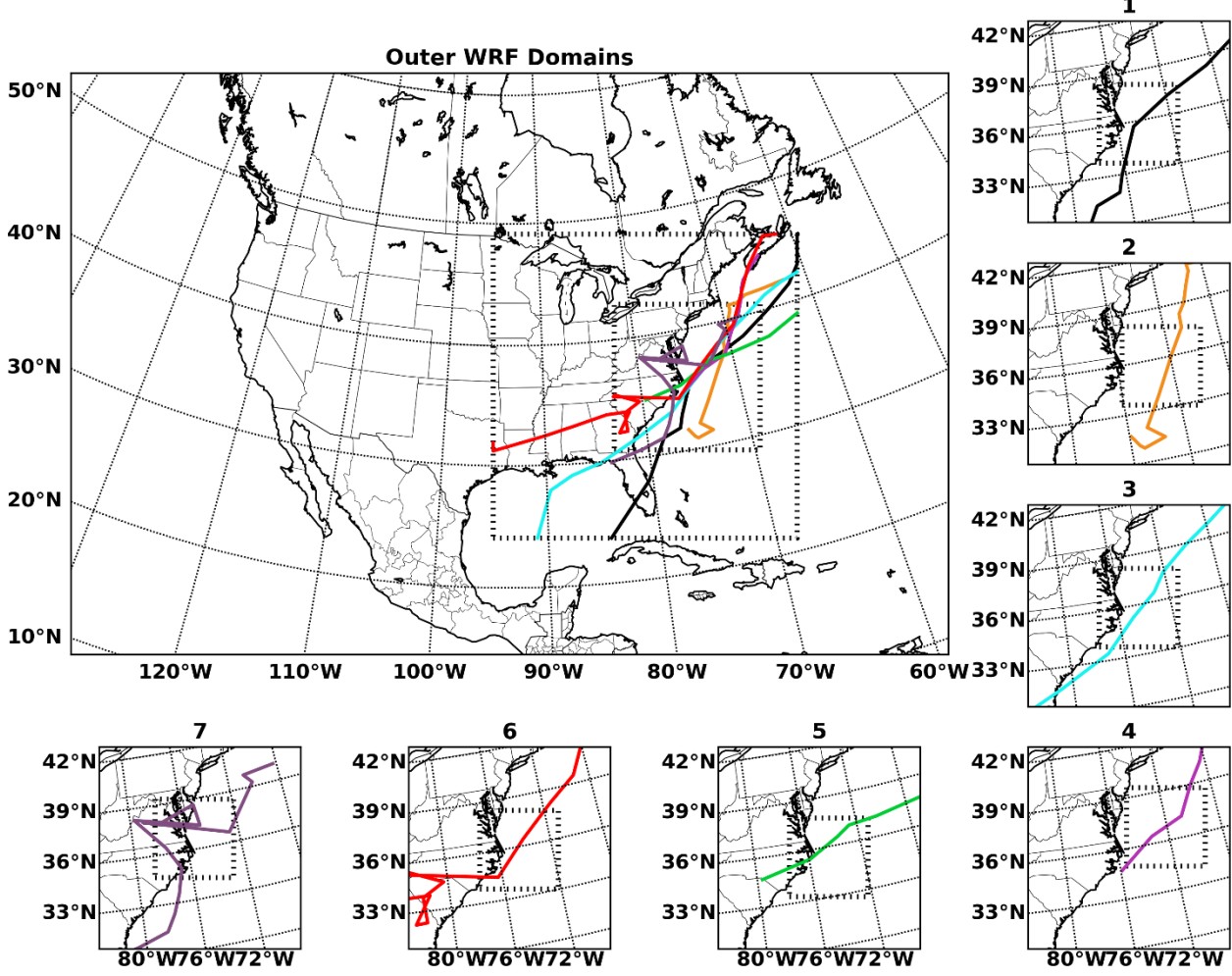

**Figure 1.** Nested WRF configuration used in simulations. The large panel shows the first 3 model domains (45-, 15-, 5- km grid spacing, respectively). The smaller panels show the location of domain 4 (1.667-km resolution) for each of the seven cases. The colored lines show the cyclone track as indicated by GMA for each nor'easter case.

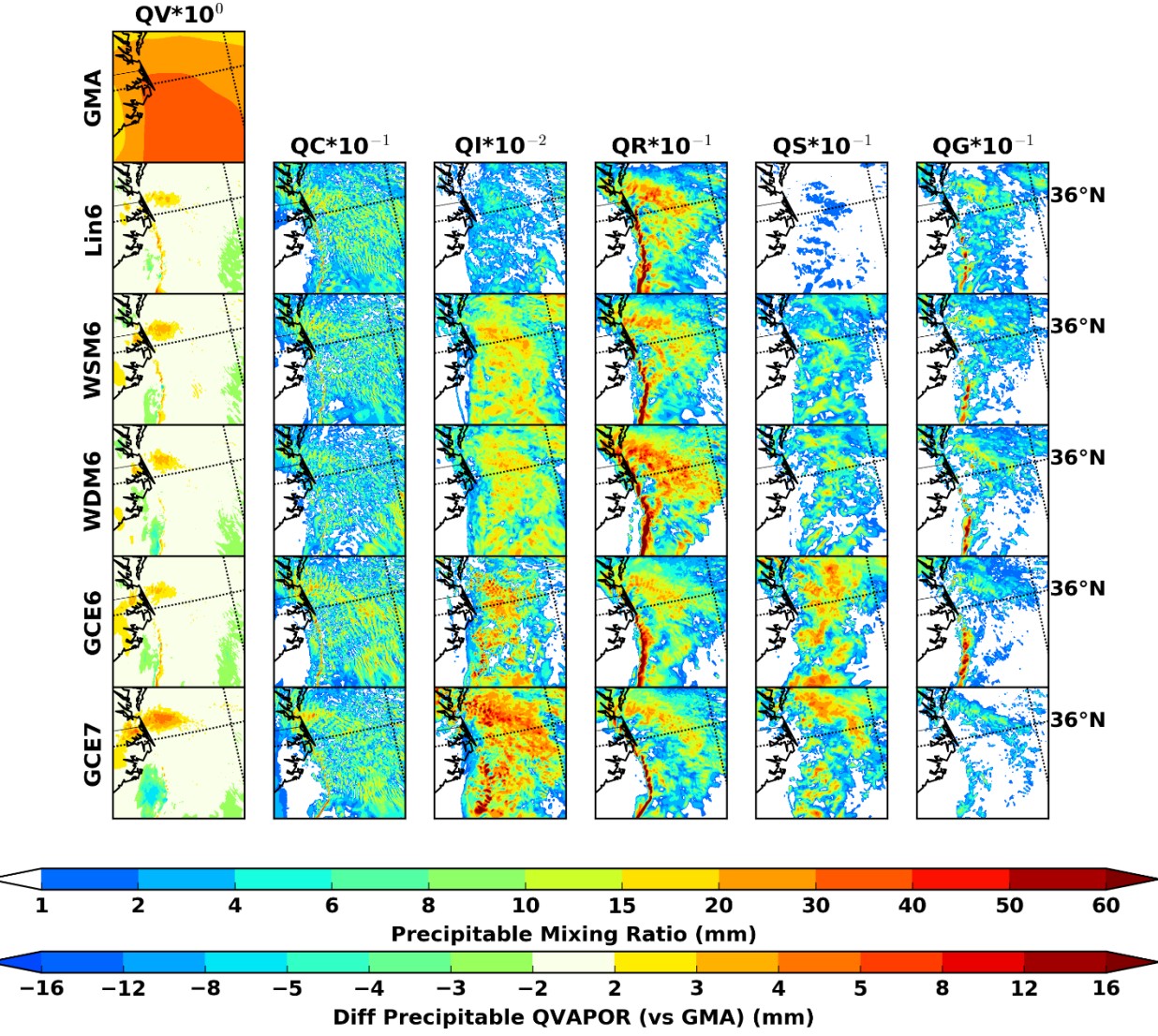

**Figure 2.** Domain 4 (1.667 km grid spacing), precipitable mixing ratios (mm) at 06 UTC 06 February 2010. Shown abbreviations for mixing ratios include: QV = water vapor, QC = cloud water, QG = graupel, QI = cloud ice, QR = rain, QS = snow.

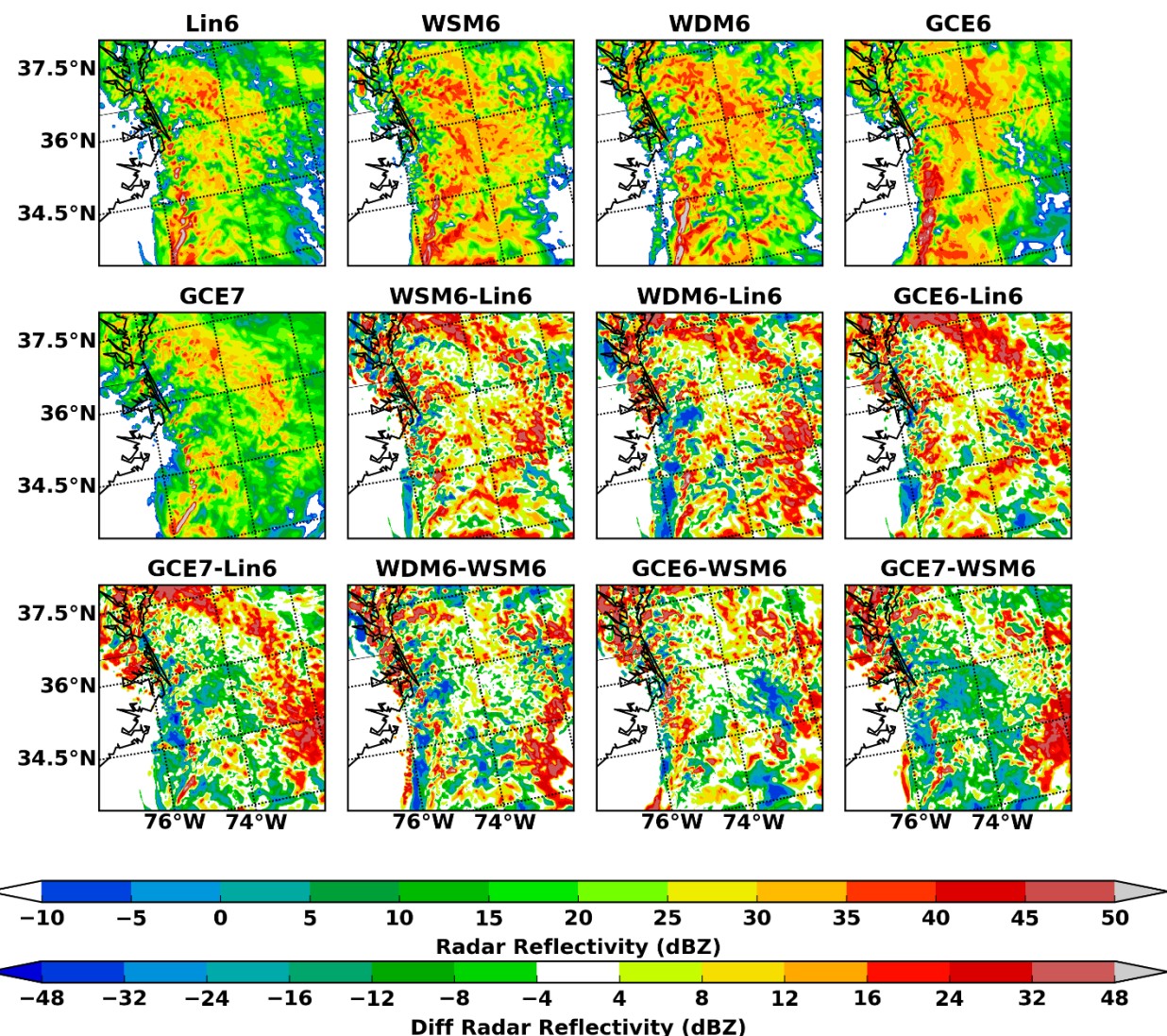

**Figure 3.** Simulated radar reflectivity (dBZ) at 4,000 m above mean sea level and their difference at the same time as Fig. 2.

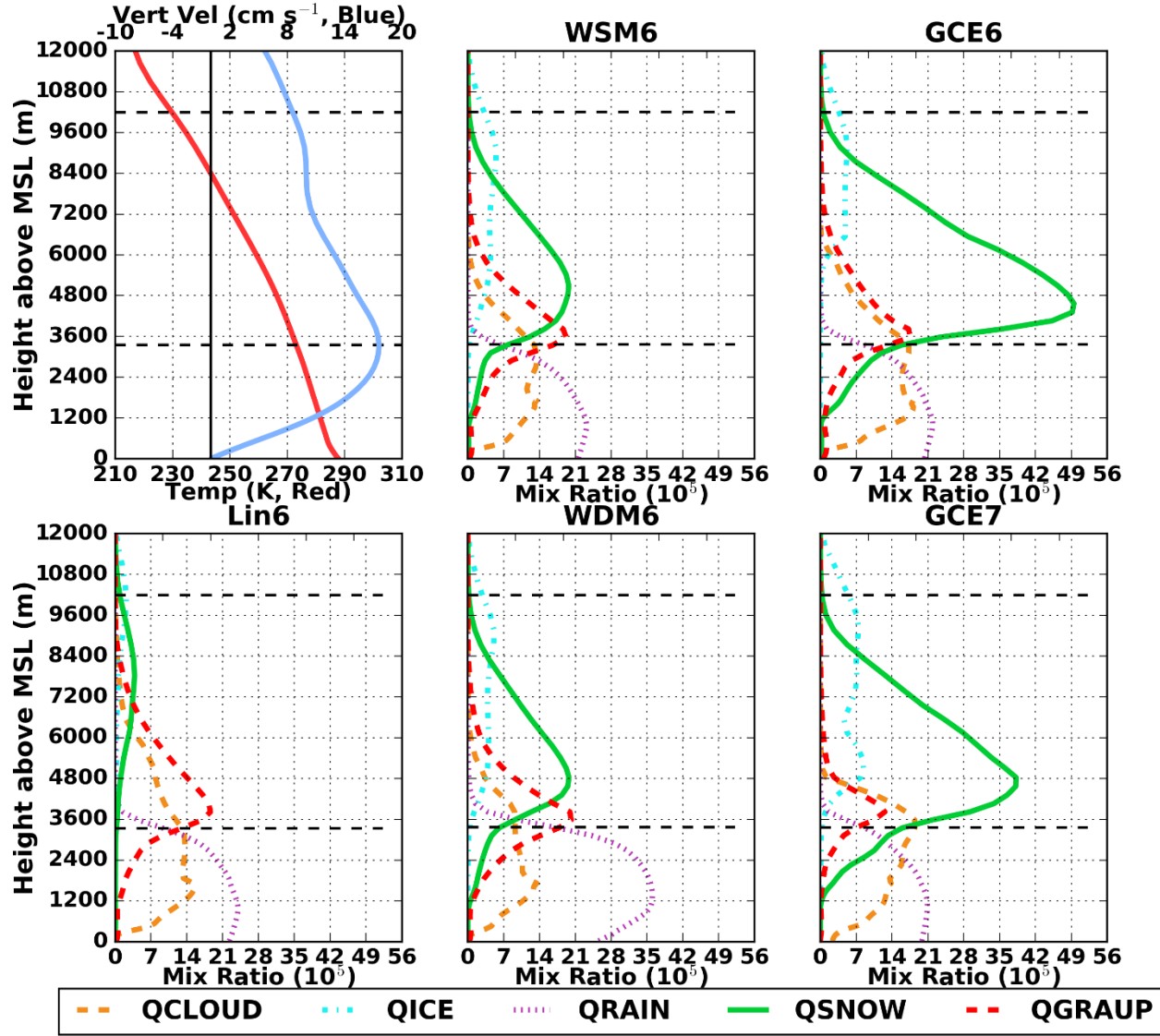

543

**Figure 4.** Domain 4-averaged (1.167-km grid spacing) mixing ratios (kg kg⁻¹), temperature (K), and vertical velocity

(cm s⁻¹) at the same time as Figs. 2 and 3. The black dashed lines denote the height above mean sea level (MSL) where

the air temperature is 0°C or -40°C. The upper-left panel shows composited and model-averaged profiles of temperature

(red line) and vertical velocity (blue). Mixing ratio species abbreviations are QCLOUD (cloud water), QGRAUP

(graupel), QICE (cloud ice), QRAIN (rain), and QSNOW (snow).

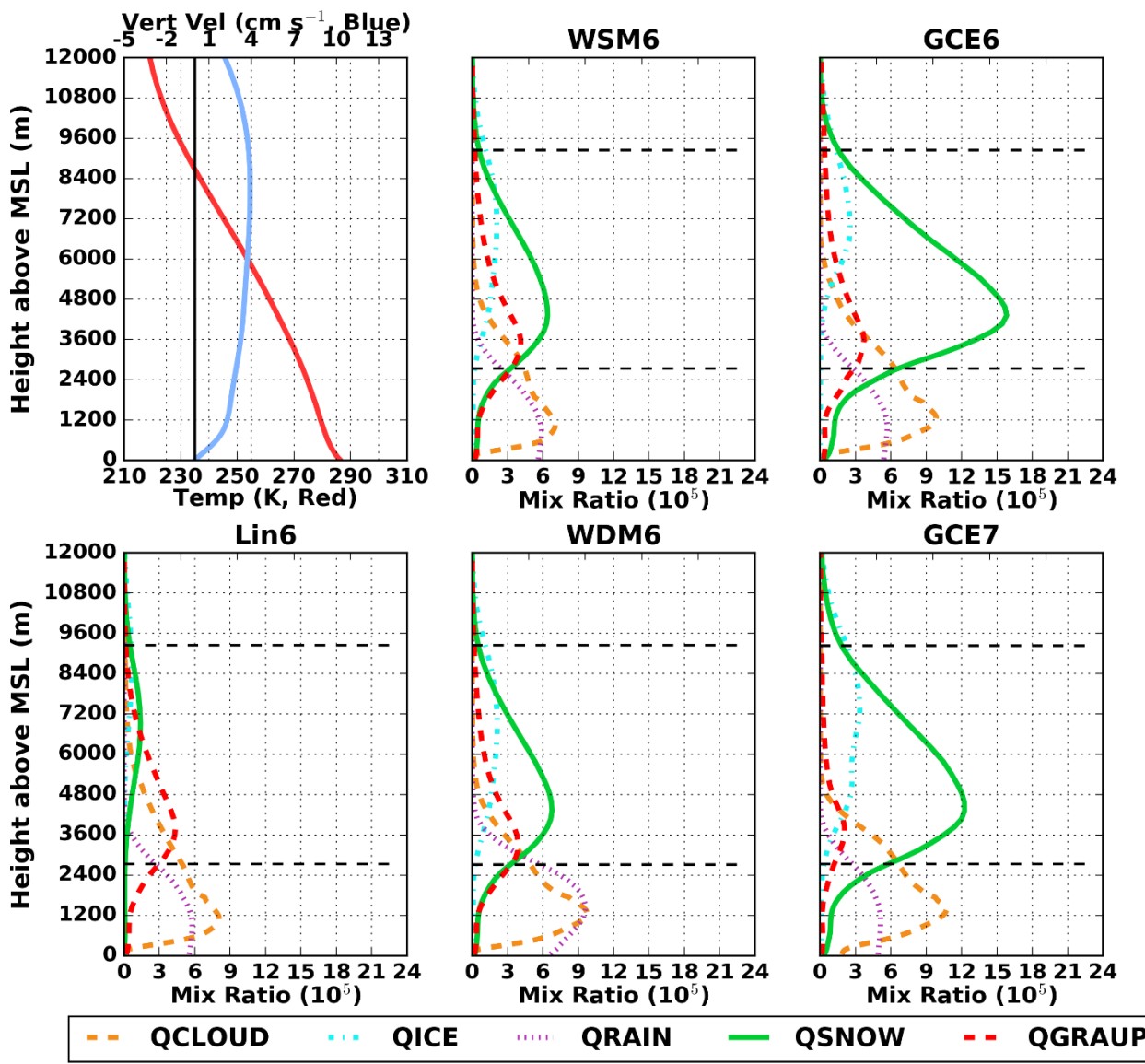

**Figure 5.** Domain 4-averaged (1.167-km grid spacing), composite mixing ratios (kg kg$^{-1}$), temperature (K), and vertical velocities (cm s$^{-1}$) composited over all seven nor'easter events. The black dashed lines denote the height above mean sea level (MSL) where the air temperature is 0°C or -40°C. The upper-left panel shows composited and model-averaged profiles of temperature (red line) and vertical velocity (blue). Mixing ratio species abbreviations are QCLOUD (cloud water), QGRAUP (graupel), QICE (cloud ice), QRAIN (rain), and QSNOW (snow).

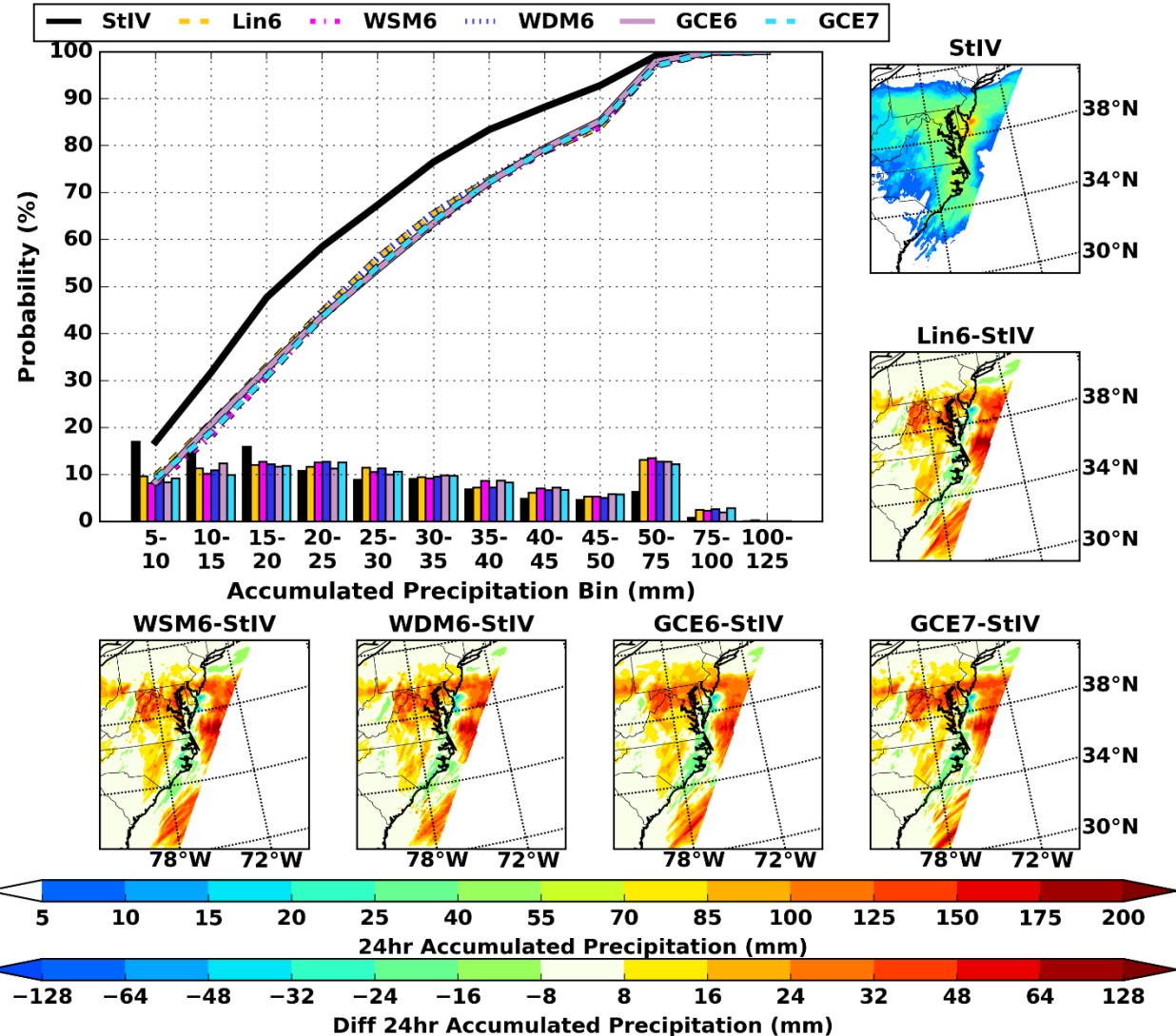

**Figure 6.** Case 5, 24-hour precipitation accumulation and their differences (mm, small panels) and corresponding probability density and cumulative distribution functions (big panel) of these same data derived from Stage IV and WRF model output. Accumulation period is from 00 UTC 06 February 2010 – 00 UTC 07 February 2010. Shown differences are model - Stage IV (StIV).

561

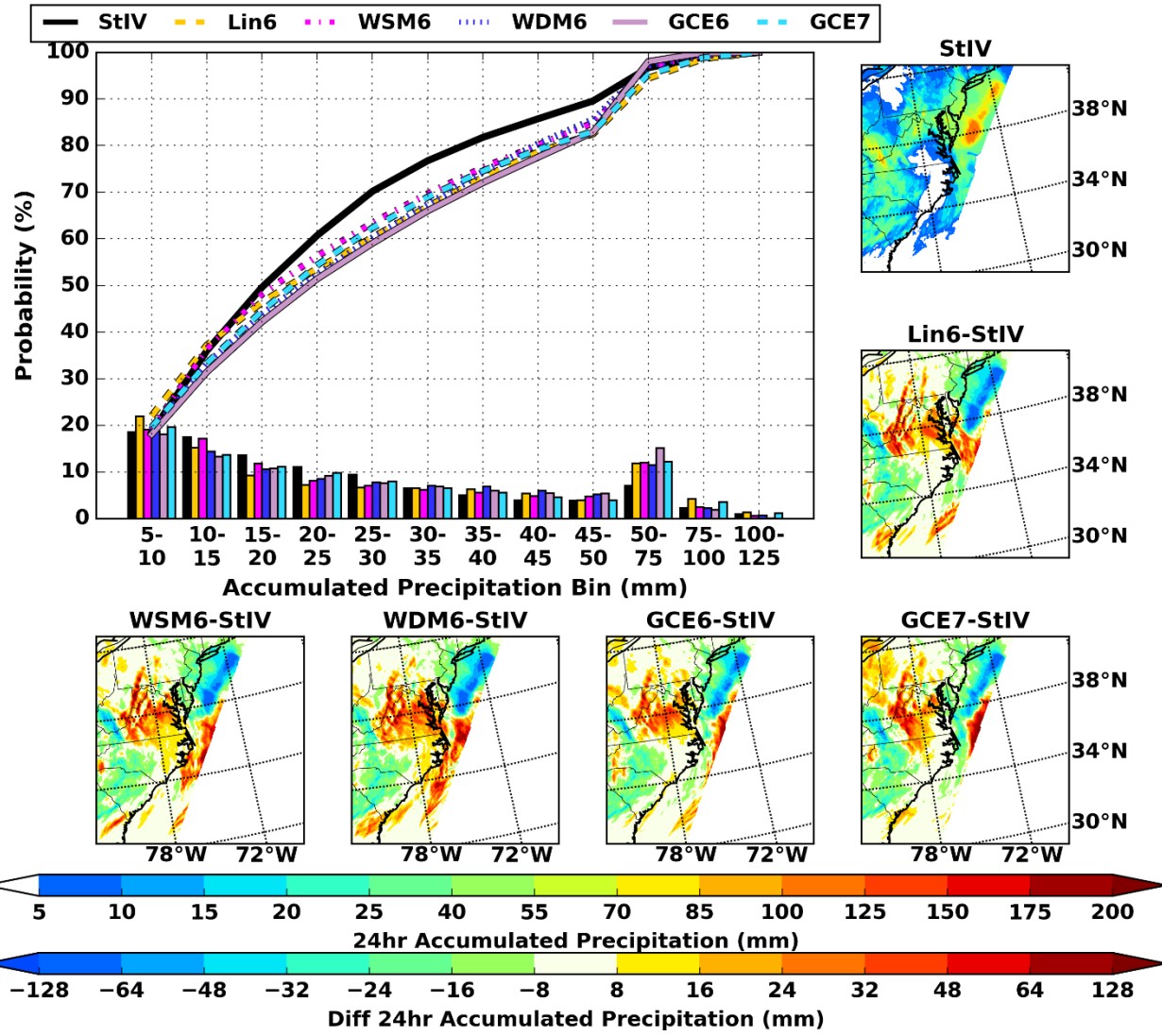

**Figure 7.** As in Fig. 6, except for Case 7. Accumulation period is from 18 UTC 12 March 2010 – 18 UTC 13 March 2010. Shown differences are model - Stage IV (StIV).

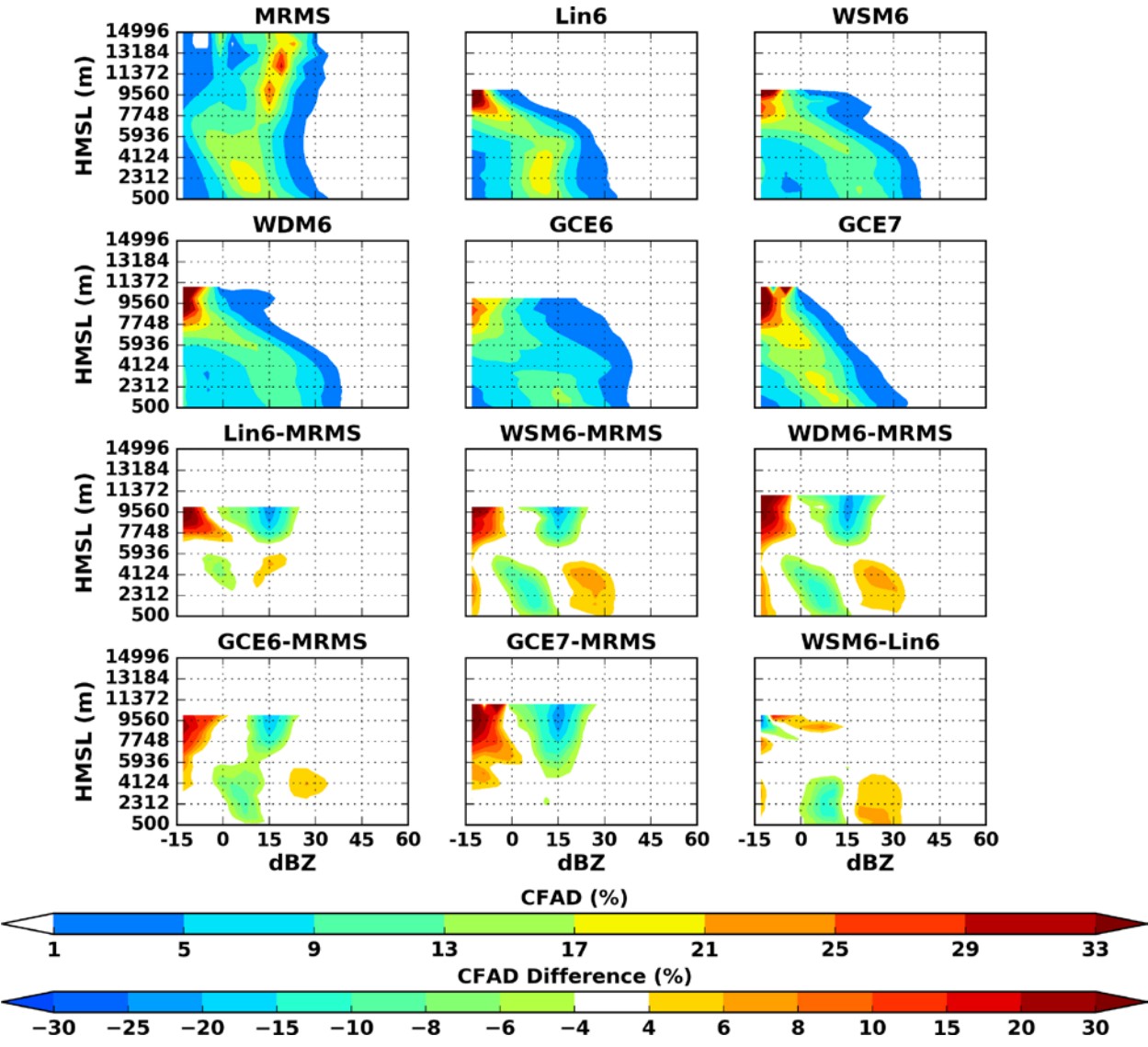

**Figure 8.** Case 4, Domain 3 (5-km grid spacing), contoured frequency with altitude diagram (CFAD) of radar reflectivity and indicated differences from Case 4 (January 2015). Data accumulation period spans 12 UTC 26 January 2015 – 12 UTC 27 January 2015 during the transit of the nor'easter through Domain 4. The y-axis shows height above mean sea level (HMSL).

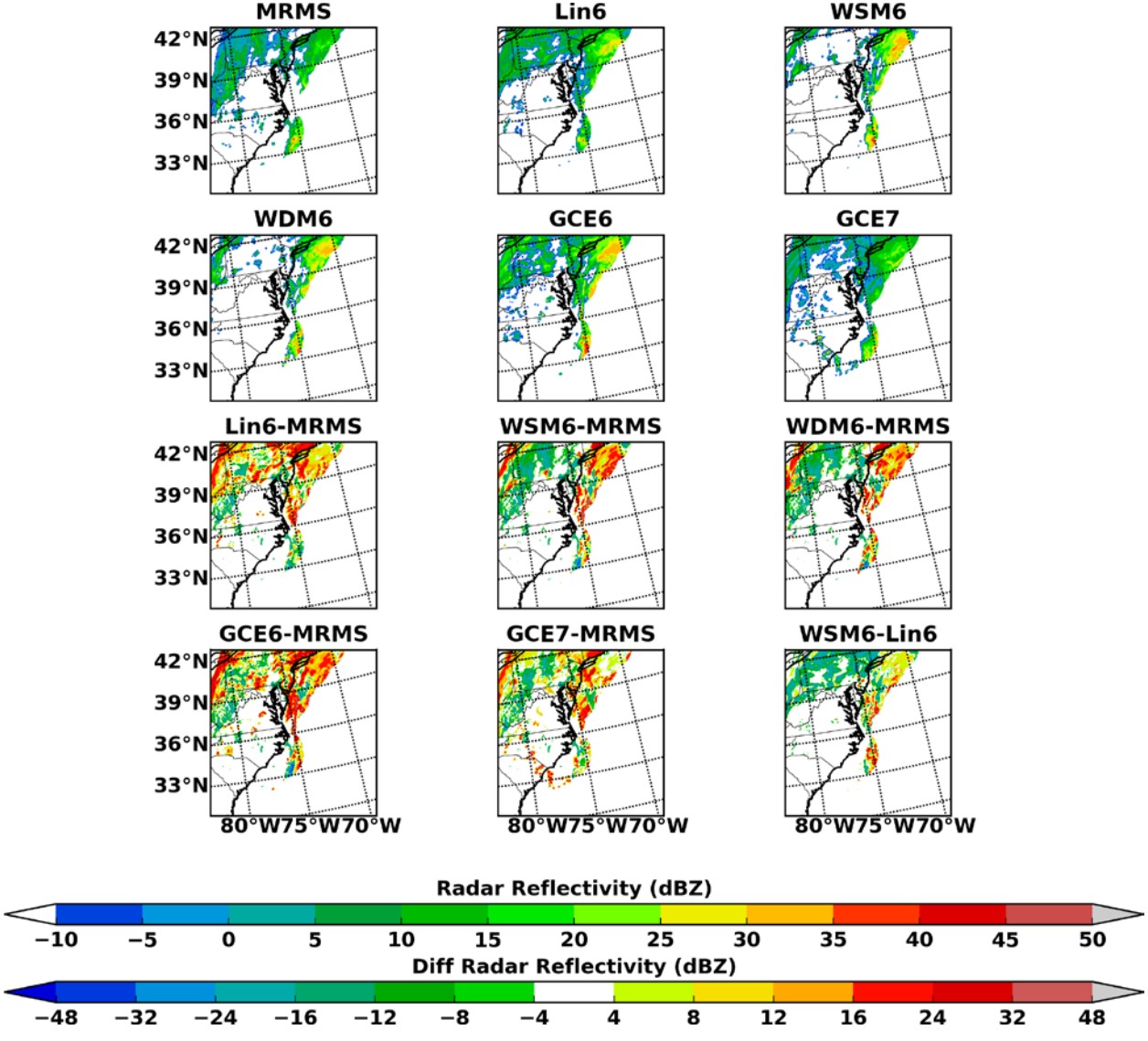

**Figure 9.** MRMS-based and WRF-simulated radar reflectivity (dBZ) at 3,000 m above sea level at 18 UTC 26 January 2015 and their differences.

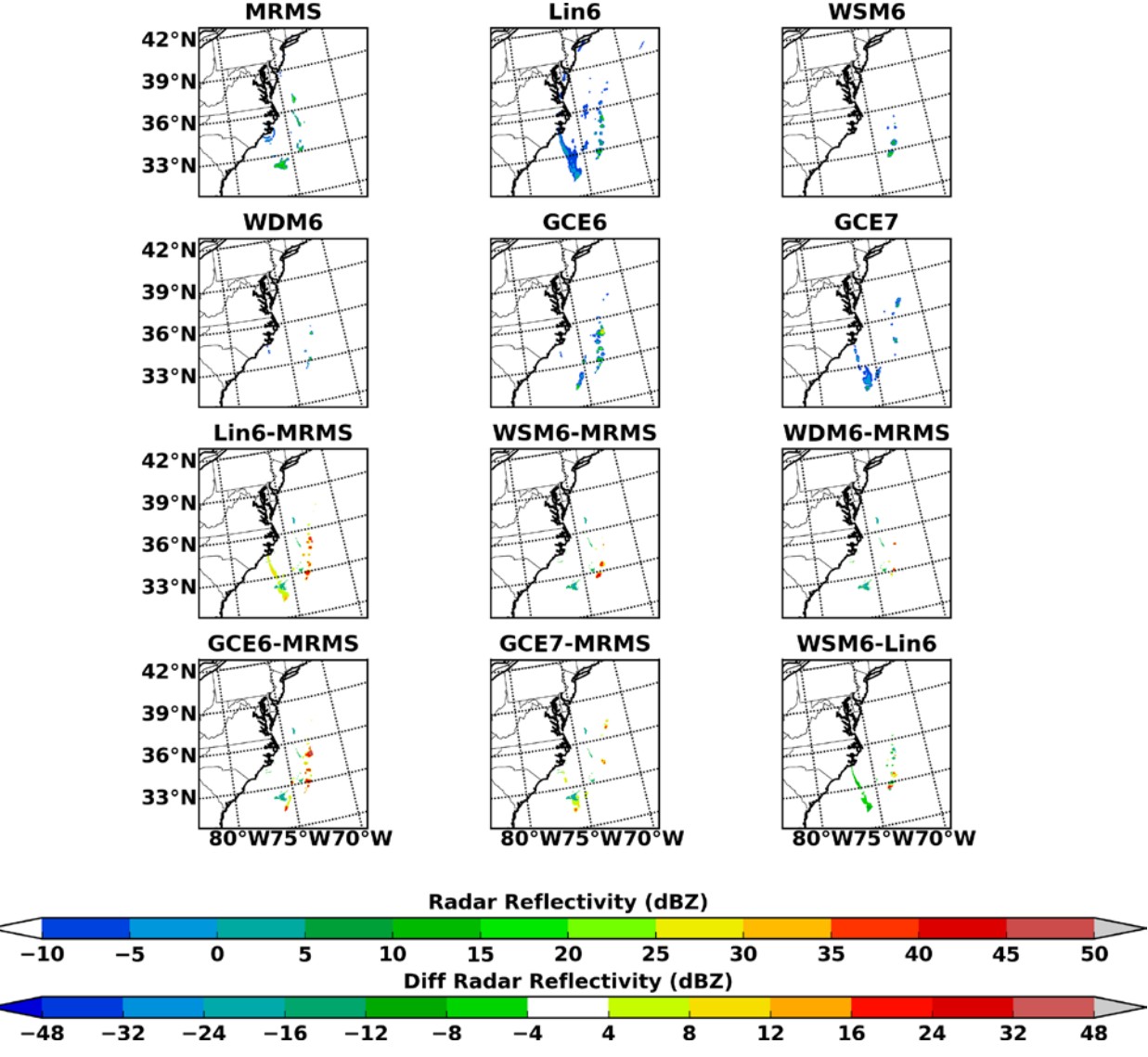

**Figure 10.** MRMS-based and WRF-simulated radar reflectivity (dBZ) at 9,000 m above sea level at 18 UTC 26 January 2015 and their differences.

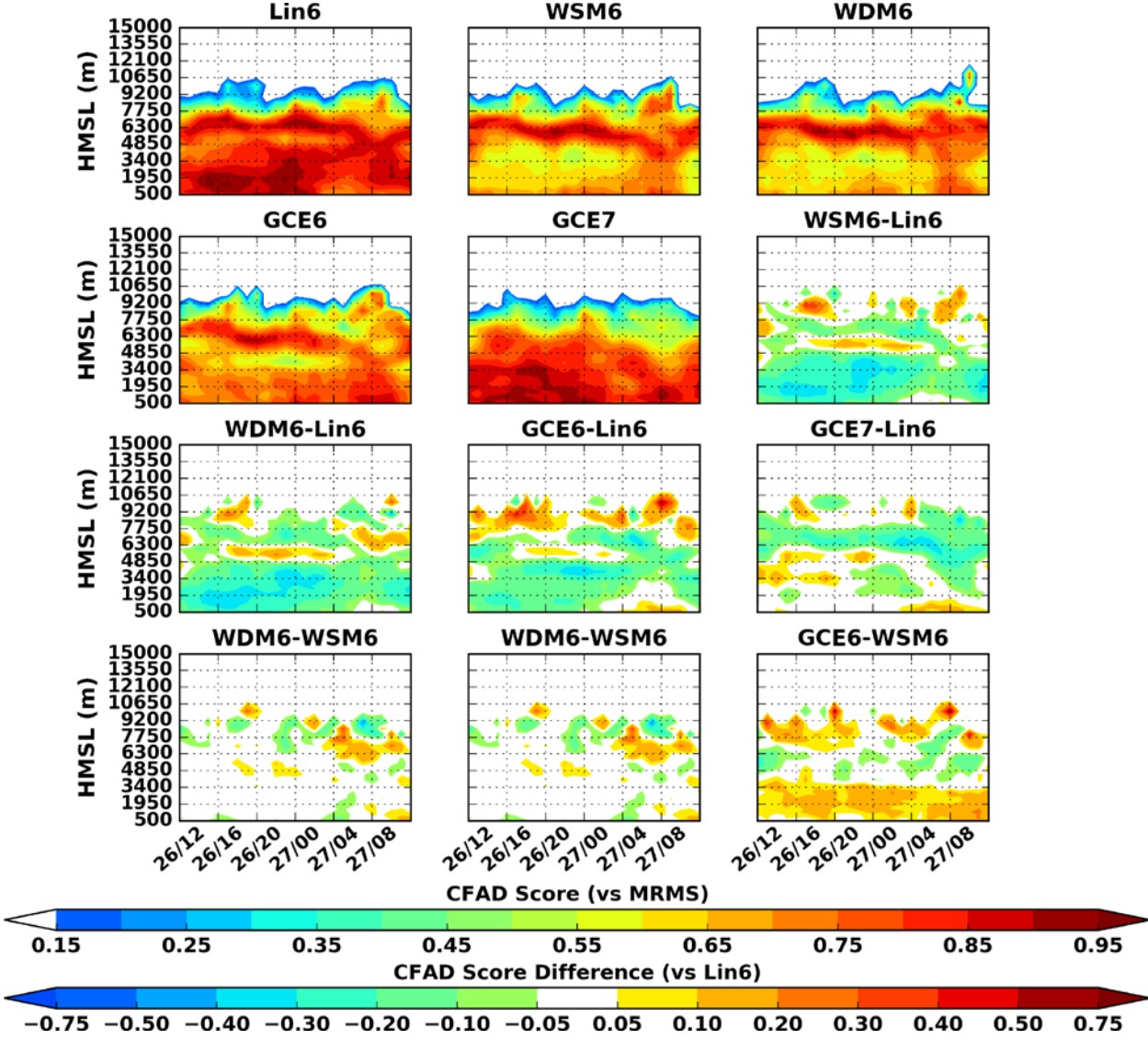

**Figure 11.** Case 4, Domain 3, (5-km grid spacing), hourly CFAD scores (See Eq. 2) of radar reflectivity and indicated differences starting at 12 UTC 26 January 2015 and ending on 12 UTC 27 January 2015. This time period corresponds to the same time period as in Figure 8. The y-axis shows height above mean sea level (HMSL) in meters.