# Peer review of "Influence of Bulk Microphysics Schemes upon Weather Research 1"

_Geoscientific Model Development, 2016_

## Referee Comment (RC1) · Anonymous Referee #1 · 27 Jul 2016

General Comments Seven Nor'easter cases were simulated with five microphysics schemes in the WRF model. The simulations were nested with four levels from 45 km down to 1.667 km. The verification focuses on Stage IV precipitation and GFS (GMA) operational analyses. The schemes are also intercompared with metrics that include the microphysical species, tracks, intensities of the central low pressure, and an energy norm. The paper has some results of interest within what is clearly a large dataset of model output. They have found that the modeled nor-easters have a stronger tendency to look more similar to each other than to the analysis, and perhaps this is not surprising because microphysics changes alone are not likely to change the motions of these systems. However, this makes comparisons with analyses harder to do, and

difficult to gain much from if the purpose is to intercompare microphysics schemes.

This brings up my main issue with the paper which is whether we can evaluate microphysics schemes against analyses such as these in a useful way. The value of the paper is more in the intercomparison of microphysics than in how they compare with analyses. Errors in the forecast are dominated by other causes, such as the initial analysis error, considering that these are initialized 72 hours ahead of the precipitation events. Perhaps initializing closer to the event would have given more accurate representations that could be compared with analyses.

I especially am not convinced that the energy norm metric has been demonstrated to be useful. The results with that show no obvious signal and are surely dominated by position errors even though the norms were centered on the model and analysis storms. The wrong track gives different sea-surface temperatures, for example, making it expected that storm properties would not be comparable.

There are also aspects of the model set-up that I would criticize. It seems that the central 1.67 km domain is at the same position for all storms, and this means that some storms pass through it while other would miss it and only be resolved in the 5 km domain. This makes for a complication in the comparisons too, especially between storms, or depending where on their track that they are verified.

Specific Comments

1. line 141. What are the perturbations relative to, the GMA analysis? This is not stated.

2. Section 3.2. It is not clear what area these results and Table 4 are for. It also seems that much of this would be in the 12 km domain where there is a cumulus scheme, and part is in domains 3 and 4 where there isn't.

3. line 208. WRF's common heritage with GFS is implied. I don't think there is much common physics heritage except for some relationship in the land-surface scheme.

What is meant here?

4. Abstract does not mention that there are seven cases and five microphysics schemes and has nothing on the energy norm. It is not adequately describing the work carried out.

5. line 234. What is meant by saturation heights?

6. line 236. cloud water? This should probably be cloud droplet number concentration?

7. line 241-246. Without knowing where the freezing level is, it is difficult to follow this discussion. How much of the cloud water is supercooled?

8. line 279. How does lack of a sedimentation term lead to low cloud ice? I thought sedimentation should reduce cloud ice extent and lifetime.

9. line 282. 'assumed water saturation'. What assumption is made about water saturation in a purely ice process?

10. Figure 7 (vapor) would have been better presented as a difference from analysis. Nothing can be seen with this plot as it is.

11. Section 3.4. It is hard to interpret what is meant by lowest energy norms and the metrics in Table 5 in general. Also make clearer what is meant by model-relative and GMA-relative norms.

12. As mentioned in the general comments, I do not think the energy norm statistics are adding anything useful to the paper. It would be better and more focused without this. There are so many factors that could make one simulation look temporarily better than another, related to timing and structure developments, that using such a high-level bulk measure as this conflates too many things to be useful in such an intercomparison.

13. line 334. Regarding the low-level jet which case is being referred to? Can it really be inferred from the v component of the energy norm that this jet is the cause? This looks highly speculative.

14. Generally I think the microphysical comparisons are the useful part of the paper and some effort has gone into explaining the results in terms of differences between the schemes. The caveat I have here is that the nesting locations make it unclear whether we are looking at the behavior at 1.67 km or 5 km, and it looks like it must be a mixture. This is a drawback of the methodology.

Technical Corrections

1. line 217. It is not stated that this is Case 4.

2. line 175 and 178. It seems that Figure 1 is not the correct one to refer to here?

3. line 200. BPMs

4. line 256. Extra 'and'.

5. line 320. Should it say WSM6?

---

## Referee Comment (RC2) · Anonymous Referee #2 · 18 Aug 2016

GENERAL COMMENTS: The paper analyzes the role of five microphysical schemes of the numerical model WRF3.6.1 upon seven cases of "nor'esters". I think that the paper is well formed and it has interesting topics.

I think that the spin-up time of 72 hours is too long for a simulation without any kind of assimilation. A test with a shorter spin up (12 hours) could be recommendable.

A microphysical comparison with observations could be useful because this topic is the main focus of the paper. Is it possible to retrieve data from radar or satellite platform? For example in http://dx.doi.org/10.1175/JAS-D-13-0107.1 the microphysical comparison has been performed using data from TRMM satellite platform.

SPECIFIC COMMENTS:

Line 133: w is the mixing ratio of rain?

Line 203: Not Fig. 4 but Fig. 5

Figs. 5-6-7: insert letters in the panel to easy the reading of section 3.
* * *

---

## Author Comment (AC1) · 27 Oct 2016

**Responses to Anonymous Referee #1 (Page 1)**

My co-authors and I wish acknowledge and thank Reviewer #1 for the time, energy, and effort applied in the detailed review of this manuscript. We do feel that a more narrow focus on microphysics and removal of the energy norm has improved upon the original manuscript and also address most if not all of the highlighted concerns.

**Responses to General Comments:**

1) "...my main issue with the paper which is whether we can evaluate microphysics schemes against analyses such as these in a useful way."

Both your comments and those of Reviewer #2 highlight this point. While we do believe that GFS analysis data can be useful for broader themes of our analysis (e.g., large-scale water vapor fields), its coarseness proves problematic was addressing specific microphysical-related questions. The revised manuscript now includes a new analysis making use of the Multi-Radar Multi-Sensor (MRMS) 3D volume data. These observation data, we argue, permit a more thorough investigation of smaller-scale impacts from the microphysics.

2) "Errors in the forecast are dominated by other causes, such as the initial analysis error, considering that these are initialized 72 hours ahead of the precipitation events. Perhaps initializing closer to the event would have given more accurate representations that could be compared with analyses."

In light of your suggestion and a similar comment from Reviewer #2, we shifted the model initialization time forward until 24 hours prior to cyclogenesis off the Mid-Atlantic United States and reran all 35 WRF model simulations. We believe that initializing 24 hours prior to cyclogenesis is ideal because it ensures each model simulation is sufficiently spun-up prior to the main cyclogenesis period and yet there are only minimal deviations (

**Fig. 1:** Nested WRF configuration for the original manuscript (left) and the revised manuscript (right). The colored lines in the right panel show the GFS model analysis storm tracks for each of the seven cases.

**Specific Comments:**

1. line 141. What are the perturbations relative to, the GMA analysis? This is not stated.

All energy norm calculations are relative to the GFS model analysis. The energy norm section has been removed from the paper.

2. Section 3.2. It is not clear what area these results and Table 4 are for. It also seems that much of this would be in the 12 km domain where there is a cumulus scheme, and part is in domains 3 and 4 where there isn't.

Table 4 was originally based upon domain 2 (15 km domain). The revised manuscript keeps the same approach, but we use domain 3 (5 km grid spacing) instead because it is of similar resolution to the Stage IV precipitation product (4 km resolution), the cumulus parameterization is turned off, and we felt that domain 4 would be over too limited an area for comparison.

3. line 208. WRF's common heritage with GFS is implied. I don't think there is much common physics heritage except for some relationship in the land-surface scheme. What is meant here?

My assumption here was based upon that simulated storm tracks between GFS and WRF would be similar given WRF's common heritage in GFS. Similar tracks would, in theory, give a greater potential of similar forecasts. My comment about this heritage is no longer necessary and it has been removed from the revised manuscript.

**Responses to Anonymous Referee #1 (Page 3)**

4. Abstract does not mention that there are seven cases and five microphysics schemes and has nothing on the energy norm. It is not adequately describing the work carried out.

Given the significant changes to the manuscript in this revision, the abstract has been updated and overhauled to more aptly describe the work conducted.

**5. line 234. What is meant by saturation heights?**

Thank you for this asking this clarification. By saturation height, I am referring to the height at which each microphysical species reached its maximum value. This value however is part of the mixing ratio profile and I think distracts from the paper. I have elected to remove this term from the revised manuscript.

**6. line 236. cloud water? This should probably be cloud droplet number concentration?**

Thank you for finding this error. "Cloud water" has been changed to "cloud droplet number concentration" in the revised manuscript.

**7. line 241-246. Without knowing where the freezing level is, it is difficult to follow this discussion. How much of the cloud water is supercooled?**

Thank you for noting this challenge to understanding the microphysical species analysis section. To provide information on how much of the cloud water is super cooled, I have modified the composite mixing ratio diagrams with two dashed black lines which indicate both the 0°C and -40°C levels.

**8. line 279. How does lack of a sedimentation term lead to low cloud ice? I thought sedimentation should reduce cloud ice extent and lifetime.**

Thank you for the noting this logic error. A quick read into the literature found a cloud resolving model study addressing this very topic. Their findings do indeed show that the impact of the sedimentation in cloud ice is to increase its conversion rate to snow and graupel and thus decreasing the amount, extent, and lifetime of cloud ice hydrometeors. I have removed the erroneous comment from the revised manuscript.

Nomura, M., Tsuboki, K. and Shinoda, T., 2012. Impact of Sedimentation of Cloud Ice on Cloud-Top Height and Precipitation Intensity of Precipitation Systems Simulated by a Cloud-Resolving Model. *気象集誌*. 第2 輯, 90(5), pp.791-806.

**9. line 282. 'assumed water saturation'. What assumption is made about water saturation in a purely ice process?**

The original GCE6 scheme generated excess super cooled cloud water at temperature below -12°C where such droplets do not often occur. Therefore water saturation was extended down to much colder temperatures which allowed cloud ice to achieve supersaturation with respect to ice and made cloud ice to snow conversion rates

**For further details please refer to page 2308 of the following reference:**

Lang, S. E., Tao, W. -K., Zeng, X., *and* Li, Y.: Reducing the biases in simulated radar reflectivities from a bulk microphysics scheme: Tropical convective systems, J. Atmos. Sci., 68, 2306–2320, *2011*.

**Responses to Anonymous Referee #1 (Page 4)**

10. Figure 7 (vapor) would have been better presented as a difference from analysis. Nothing can be seen with this plot as it is.

11. Section 3.4. It is hard to interpret what is meant by lowest energy norms and the metrics in Table 5 in general. Also make clearer what is meant by model-relative and GMA-relative norms.

"GMA-relative" denotes diagnosing the simulated environment within a 600-km wide box centered on the GMA-indicated cyclone center in both GMA and each WRF simulation. "Model-relative" uses the same box, but centers it on the cyclone center determined from each individual model simulation. The energy norm analysis is no longer part of the manuscript.

12. As mentioned in the general comments, I do not think the energy norm statistics are adding anything useful to the paper. It would be better and more focused without this. There are so many factors that could make one simulation look temporarily better than another, related to timing and structure developments, that using such a high-level bulk measure as this conflates too many things to be useful in such an intercomparison.

While we do see some value in the energy norm results with respect to diagnosing which dynamical fields are responsible for observed error, we agree that in context of a microphysics- focused paper this metric is not sensitive enough to be of use. Pending the suggestion of both reviewers, this section has been redacted from the revised manuscript.

13. line 334. Regarding the low-level jet which case is being referred to? Can it really be inferred from the v component of the energy norm that this jet is the cause? This looks highly speculative.

We agree with the reviewer's viewpoint that the energy norm by itself could be considered speculative for Case 7. Our decision to not include a figure of 850-hPa winds (See Figure 2 below) in the original manuscript was made on the assumption that presence of the cyclone center, the small size of the model domain, and a bump in the u and v energy norm components at 850-hPa would be sufficient circumstantial evidence to support our claim without the need for an additional figure. In the revised manuscript, the energy norm section has been removed from the paper.

---

## Author Comment (AC2) · 27 Oct 2016

**Responses to Anonymous Referee #2 (Page 1)**

My co-authors and I wish to thank Reviewer #2 for their time and consideration in reviewing this manuscript. Many comments are consistent with those of Reviewer #1 and have been incorporated into the revised manuscript.

**General Comments**

*1)  I think that the spin-up time of 72 hours is too long for a simulation without any kind of assimilation. A test with a shorter spin up (12 hours) could be recommendable*

In light of your suggestion and a similar comment from Reviewer #1, we shifted the model initialization time forward until 24 hours prior to cyclogenesis off the Mid-Atlantic United States and re-ran all 35 WRF model simulations. We set our start time 24 hours beforehand because simulated radar reflectivity fields still appeared slightly "blooby" up through 9-10 hours. Starting the model simulations 24 hours before primary cyclogenesis allowed for full development of simulated radar reflectivity structures and WRF-GMA track differences tended to be modest (<50 km).

*2)  "A microphysical comparison with observations could be useful because this topic is the main focus of the paper. Is it possible to retrieve data from radar or satellite platform"*

Thanks to your suggestion, we have given this revised paper more of a microphysics-style focus. I looked both into TRMM and CloudSat 2C-Ice products. TRMM offers a wide range radar observations but its orbital inclination is 35 degree (http://disc.sci.gsfc.nasa.gov/precipitation/additional/instruments/trmm_instr.html), which limits its usefulness when only half my analysis domains falls equatorward of 35°N. CloudSAT does provide profiles cloud ice, which my collegue used in a recent paper on global cloud species. It narrow swath range (see Figure 3) made getting a consistent "hit" on a nor'easter challenging.

[Figure]

**Fig. 3:** CloudSAT orbital overpass sample from 2012.

I did find success with the Multi-Radar Multi-Sensor product from National Oceanagrahic and Atmospheric Association (NOAA), which provides hourly gridded 3D volume scans at 1-hour intervals (See Figure 4). Similar to StageIV, MRMS data only covers part of domain 4 in many of the seven cases, but the results thus far have been reasonable and useful.

**Responses to Anonymous Referee #2 (Page 2)**

[Figure]

**Fig. 4:** MRMS coverage area (everywhere with colors).

**Specific Comments:**

*1) Line 133: w is the mixing ratio of rain?*

Although 'w' is often used in meteorology to denote mixing ratio, it represents vertical velocity in the energy norm equation. Instead, this formula uses 'q' to represent mixing ratio. With the removal of the energy norm from the paper's results this particular comment is no longer valid.

*2) Line 203: Not Fig. 4 but Fig. 5*

Thank you for catching the typo. I have corrected the manuscript to refer to Fig. 5.

*3) Figs. 5-6-7: insert letters in the panel to easy the reading of section 3.*

While I will not dispute that Figs. 5-7 do attempt to show much data. In an earlier form of this paper, I actually tried putting letters into the panels, but these letters were difficult to place without blocking or interfering with the displayed data. I thank you for the suggestion, but I have decided to keep my "Microsoft Excel-like" approach to plot labelling.

---

## Author Response (AR2)

**Responses to Anonymous Referee #1 (Page 1)**

My co-authors and I wish acknowledge and thank Reviewer #1 for the time, energy, and effort applied in the detailed review of this manuscript. We do feel that a more narrow focus on microphysics and removal of the energy norm has improved upon the original manuscript and also address most if not all of the highlighted concerns.

**Responses to General Comments:**

1) *"…my main issue with the paper which is whether we can evaluate microphysics schemes against analyses such as these in a useful way."*

Both your comments and those of Reviewer #2 highlight this point. While we do believe that GFS analysis data can be useful for broader themes of our analysis (e.g., large-scale water vapor fields), its coarseness proves problematic was addressing specific microphysical-related questions. The revised manuscript now includes a new analysis making use of the Multi-Radar Multi-Sensor (MRMS) 3D volume data. These observation data, we argue, permit a more thorough investigation of smaller-scale impacts from the microphysics.

2) *"Errors in the forecast are dominated by other causes, such as the initial analysis error, considering that these are initialized 72 hours ahead of the precipitation events. Perhaps initializing closer to the event would have given more accurate representations that could be compared with analyses."*

In light of your suggestion and a similar comment from Reviewer #2, we shifted the model initialization time forward until 24 hours prior to cyclogenesis off the Mid-Atlantic United States and re-ran all 35 WRF model simulations. We believe that initializing 24 hours prior to cyclogenesis is ideal because it ensures each model simulation is sufficiently spun-up prior to the main cyclogenesis period and yet there are only minimal deviations (< 50 km) between WRF simulations and the GFS model analysis storm tracks.

3) *"I especially am not convinced that the energy norm metric has been demonstrated to be useful."*

We concur and agree that the energy norm, although useful, is not the most effective vehicle by which to evaluate microphysical-related simulation errors. Thus the energy norm would be more apt in a more general, bulk analysis of nor'easters where a focus on large-scale players are key. Due to our shift in model initialization time (see #2 above) and our shift to focus on microphysics (see #1 above), the energy norm analysis has been redacted from the revised manuscript.

4) *"There are also aspects of the model set-up that I would criticize. It seems that the central 1.67 km domain is at the same position for all storms, and this means that some storms pass through it while other would miss it and only be resolved in the 5 km domain"*

The WRF model domain positions were fixed for all nor'easter cases. This lead to a situation WRF-simulated nor'easters in cases 1 and 4 either missed or never fully entered the 1.667 km model grid (Domain 4) as the reviewer hypothesized. We have since increased the sizes of the 5 km and 1.167 (Domains 3 and 4, respectively) by 50%, shifted domain 3 southward, and tailored the location of domain 4 for all seven nor'easter events. To physically demonstrate these changes, Figure 1 shows our original and new WRF model configuration. All 35 model simulations were re-run and reanalyzed accordingly. As can be seen below, each model analysis track moves through the center of each respective domain 4.

[Figure]

**Fig. 1:** Nested WRF configuration for the original manuscript (left) and the revised manuscript (right).
The colored lines in the right panel show the GFS model analysis storm tracks for each of the seven cases.
**Specific Comments:**
*1. line 141. What are the perturbations relative to, the GMA analysis? This is not stated.*
All energy norm calculations are relative to the GFS model analysis. The energy norm section has
been removed from the paper.
*2. Section 3.2. It is not clear what area these results and Table 4 are for. It also seems*
*that much of this would be in the 12 km domain where there is a cumulus scheme, and*
*part is in domains 3 and 4 where there isn't.*
Table 4 was originally based upon domain 2 (15 km domain). The revised manuscript keeps the
same approach, but we use domain 3 (5 km grid spacing) instead because it is of similar resolution to the
Stage IV precipitation product (4 km resolution), the cumulus parameterization is turned off, and we felt
that domain 4 would be over too limited an area for comparison.
*3. line 208. WRF's common heritage with GFS is implied. I don't think there is much*
*common physics heritage except for some relationship in the land-surface scheme. What is meant here?*
My assumption here was based upon that simulated storm tracks between GFS and WRF would
be similar given WRF's common heritage in GFS. Similar tracks would, in theory, give a greater potential
of similar forecasts. My comment about this heritage is no longer necessary and it has been removed from
the revised manuscript.

**Responses to Anonymous Referee #1 (Page 3)**

*4. Abstract does not mention that there are seven cases and five microphysics schemes and has nothing on the energy norm. It is not adequately describing the work carried out.*

Given the significant changes to the manuscript in this revision, the abstract has been updated and overhauled to more aptly describe the work conducted.

*5. line 234. What is meant by saturation heights?*

Thank you for this asking this clarification. By saturation height, I am referring to the height at which each microphysical species reached its maximum value. This value however is part of the mixing ratio profile and I think distracts from the paper. I have elected to remove this term from the revised manuscript.

*6. line 236. cloud water? This should probably be cloud droplet number concentration?*

Thank you for finding this error. "Cloud water" has been changed to "cloud droplet number concentration" in the revised manuscript.

*7. line 241-246. Without knowing where the freezing level is, it is difficult to follow this discussion. How much of the cloud water is supercooled?*

Thank you for noting this challenge to understanding the microphysical species analysis section. To provide information on how much of the cloud water is super cooled, I have modified the composite mixing ratio diagrams with two dashed black lines which indicate both the 0°C and -40°C levels.

*8. line 279. How does lack of a sedimentation term lead to low cloud ice? I thought sedimentation should reduce cloud ice extent and lifetime.*

Thank you for the noting this logic error. A quick read into the literature found a cloud resolving model study addressing this very topic. Their findings do indeed show that the impact of the sedimentation in cloud ice is to increase its conversion rate to snow and graupel and thus decreasing the amount, extent, and lifetime of cloud ice hydrometeors. I have removed the erroneous comment from the revised manuscript.

Nomura, M., Tsuboki, K. and Shinoda, T., 2012. Impact of Sedimentation of Cloud Ice on Cloud-Top Height and Precipitation Intensity of Precipitation Systems Simulated by a Cloud-Resolving Model. 気象集誌. 第2 輯, *90*(5), pp.791-806.

*9. line 282. 'assumed water saturation'. What assumption is made about water saturation in a purely ice process?*

The original GCE6 scheme generated excess super cooled cloud water at temperature below -12°C where such droplets do not often occur. Therefore water saturation was extended down to much colder temperatures which allowed cloud ice to achieve supersaturation with respect to ice and made cloud ice to snow conversion rates

**For further details please refer to page 2308 of the following reference:**
Lang, S. E., Tao, W. -K., Zeng, X., *and* Li, Y.: Reducing the biases in simulated radar reflectivities from a bulk microphysics scheme: Tropical convective systems, J. Atmos. Sci., 68, 2306–2320, *2011.*

**Responses to Anonymous Referee #1 (Page 4)**

*10. Figure 7 (vapor) would have been better presented as a difference from analysis.*
*Nothing can be seen with this plot as it is.*

Thank you for the suggestion. In new manuscript, this diagram (now Figure 2) has been updated to show the difference in water vapor.

*11. Section 3.4. It is hard to interpret what is meant by lowest energy norms and the*
*metrics in Table 5 in general. Also make clearer what is meant by model-relative and*
*GMA-relative norms.*

"GMA-relative" denotes diagnosing the simulated environment within a 600-km wide box centered on the GMA-indicated cyclone center in both GMA and each WRF simulation. "Model-relative" uses the same box, but centers it on the cyclone center determined from each individual model simulation. The energy norm analysis is no longer part of the manuscript.

*12. As mentioned in the general comments, I do not think the energy norm statistics are*
*adding anything useful to the paper. It would be better and more focused without this.*
*There are so many factors that could make one simulation look temporarily better than*
*another, related to timing and structure developments, that using such a high-level bulk*
*measure as this conflates too many things to be useful in such an intercomparison.*

While we do see some value in the energy norm results with respect to diagnosing which dynamical fields are responsible for observed error, we agree that in context of a microphysics- focused paper this metric is not sensitive enough to be of use. Pending the suggestion of both reviewers, this section has been redacted from the revised manuscript.

*13. line 334. Regarding the low-level jet which case is being referred to? Can it really*
*be inferred from the v component of the energy norm that this jet is the cause? This*
*looks highly speculative.*

We agree with the reviewer's viewpoint that the energy norm by itself could be considered speculative for Case 7. Our decision to not include a figure of 850-hPa winds (See Figure 2 below) in the original manuscript was made on the assumption that presence of the cyclone center, the small size of the model domain, and a bump in the u and v energy norm components at 850-hPa would be sufficient circumstantial evidence to support our claim without the need for an additional figure. In the revised manuscript, the energy norm section has been removed from the paper.

[Figure]

**Fig. 2:** 850-hPa wind speed (fills, m s-1) and sea-level pressure (contours, hPa) on 13 March 2010 at 18 UTC (Case 7).

**Responses to Anonymous Referee #2 (Page 1)**

My co-authors and I wish to thank Reviewer #2 for their time and consideration in reviewing this manuscript. Many comments are consistent with those of Reviewer #1 and have been incorporated into the revised manuscript.

**General Comments**

1) *I think that the spin-up time of 72 hours is too long for a simulation without any kind of assimilation. A test with a shorter spin up (12 hours) could be recommendable*

In light of your suggestion and a similar comment from Reviewer #1, we shifted the model initialization time forward until 24 hours prior to cyclogenesis off the Mid-Atlantic United States and re-ran all 35 WRF model simulations. We set our start time 24 hours beforehand because simulated radar reflectivity fields still appeared slightly "blooby" up through 9-10 hours. Starting the model simulations 24 hours before primary cyclogenesis allowed for full development of simulated radar reflectivity structures and WRF-GMA track differences tended to be modest (<50 km).

2) *"A microphysical comparison with observations could be useful because this topic is the main focus of the paper. Is it possible to retrieve data from radar or satellite platform"*

Thanks to your suggestion, we have given this revised paper more of a microphysics-style focus. I looked both into TRMM and CloudSat 2C-Ice products. TRMM offers a wide range radar observations but its orbital inclination is 35 degree (http://disc.sci.gsfc.nasa.gov/precipitation/additional/instruments/trmm_instr.html), which limits its usefulness when only half my analysis domains falls equatorward of 35°N. CloudSAT does provide profiles cloud ice, which my colleague used in a recent paper on global cloud species. It narrow swath range (see Figure 3) made getting a consistent "hit" on a nor'easter challenging.

[Figure]

**Fig. 3:** CloudSAT orbital overpass sample from 2012.

I did find success with the Multi-Radar Multi-Sensor product from National Oceanagrahic and Atmospheric Association (NOAA), which provides hourly gridded 3D volume scans at 1-hour intervals (See Figure 4). Similar to StageIV, MRMS data only covers part of domain 4 in many of the seven cases, but the results thus far have been reasonable and useful.

[Figure]

**Fig. 4:** MRMS coverage area (everywhere with colors).
**Specific Comments:**
*1) Line 133: w is the mixing ratio of rain?*
Although 'w' is often used in meteorology to denote mixing ratio, it represents vertical velocity in the
energy norm equation. Instead, this formula uses 'q' to represent mixing ratio. With the removal of the
energy norm from the paper's results this particular comment is no longer valid.
*2) Line 203: Not Fig. 4 but Fig. 5*
Thank you for catching the typo. I have corrected the manuscript to refer to Fig. 5.
*3) Figs. 5-6-7: insert letters in the panel to easy the reading of section 3.*
While I will not dispute that Figs. 5-7 do attempt to show much data. In an earlier form of this paper,
I actually tried putting letters into the panels, but these letters were difficult to place without blocking or
interfering with the displayed data. I thank you for the suggestion, but I have decided to keep my
"Microsoft Excel-like" approach to plot labelling.

**Responses to Editor (Page 1)**

Note to editor: Apologies for taking so long with these technical corrections, the end of the year is always a busy time and my co-authors were hard to pin down. While having a paper returned with "major revisions" is never a good feeling, we were glad to have the extra time because we were able to make your suggested corrections, do many other tweaks to the paper (tables, figures, wording, etc.), and even do a little more analysis. We hope the end product meets your satisfaction and that our efforts here (with luck) will be viewed favorably by the reviewers. A summary of the highlighted corrections and our response is below

Editor highlighted corrections (Corrections or changes are noted in bold):

1) *l158: ITS mixing ratio*

   **Original:** We exclude hail from our analysis because it is unique to GCE7 and it mixing ratio values are an order of magnitude smaller than other species.

   **Modified:** We exclude hail from our analysis because it is unique to GCE7 and **its** mixing ratio values are an order of magnitude smaller than other species.

2) *l168: will BE explained*

   **Original:** …is due to identifiable trends within the underlying assumptions made by BMPSs and will explained in more detail below

   **Modified:** …is due to identifiable trends within the underlying assumptions made by BMPSs and will **be** explained in more detail below

3) *l207: ITS*
   **Original:** GCE7 is in many ways at opposition to Lin6, where it simulations generate the most snow, …
   **Modified:** GCE7 is in many ways at opposition to Lin6, where **its** simulations generate the most snow, …

4) *l235: turned on*

   **Original:** WRF precipitation is generated from its microphysics and cumulus parameterization; the latter is turned for Domains 3 (5 km grid spacing) and 4 (1.667-km grid spacing)

   **Modified:** WRF precipitation is generated from its microphysics and cumulus parameterization; the latter is turned **on** for Domains 3 (5 km grid spacing) and 4 (1.667-km grid spacing)

5) l255: IN

   **Original:** As illustrated Figs. 6 and 7, all WRF simulations tended to generate similar coverage to Stage IV, …

   **Modified:** As illustrated **in** Figs. 6 and 7, all WRF simulations tended to generate similar coverage to Stage IV, …

**Responses to Editor (Page 2)**

*6)  l272-273: ??*

No direct comment is given. I am assuming that the statement or phrasing here is confusing. In the revised manuscript, I no longer reference all the CFAD and CFAD-related figures in the first paragraph. Instead, I now describe the CFAD plot (Fig. 8) generally and then introduce the follow-on figures (Figs. 9-11) as needed. For reference the 3,000 m and 9,000 m heights were selected because they represented levels were the BMPSs varied (3,000 m) or where errors with the MRMS data product filtering (9,000 m) could be highlighted.

7)  l305: ??
Similar to 6, I am assuming this line is confusing and too long. I have revised the sentence and boiled it down to more exact details.

**Original:** WRF-Stage IV accumulated precipitation comparisons reveal WRF demonstrate that although WRF generates precipitation fields of similar coverage to Stage IV precipitation intensities tended to be higher than observations and resulting in low to moderate (0.217–0.414) threat scores with WDM6 demonstrating marginally better forecast skill than its single-moment counterparts.

**Modified:** WRF-simulated precipitation fields exhibit similar coverage but trended towards higher precipitation amounts relative to Stage IV observations resulting in low-to-moderate threat scores (0.217–0.414).

8)  l310:

Similar to 6 and 7, I am assuming this section of the manuscript was confusing or unclear. In the latest revision of the results, we were able to pull out more details about the Lin6 vs GCE7 comparison. By focusing more on GCE7 and Lin6 in our description, we think this helps make the conclusion here clearer to the reader.

**Original:** Finally, MRMS-based CFAD and CFAD scores show Lin6 and GCE7 to be notably better than GCE6, WSM6 and WDM6 in the lower troposphere, with GCE7 being the only BMPS scheme to produce the narrow core of maximum frequencies below10 dBZ due to its temperature and mixing ratio dependent aggregation and new snow map. Above 5,000 m GCE7 however becomes less skilled the combination of smaller hydrometers and entrainment reduced it cloud top height relative to other BMPSs.

**Modified:** Finally, MRMS-based contoured frequency with altitude diagrams (CFADs) and CFAD scores show Lin6 and GCE7 to perform the best in the lower half of the troposphere (below 6,300 m AMSL), where GCE7 most realistically reproduced the maximum frequency core between 5 and 15 dBZ due to its temperature and mixing ratio dependent aggregation and new snow size mapping. However, the overly large growth of graupel via its dry collection of snow suggests that Lin6 obtains high CFAD scores from a less realistic solution than GCE7. Above 6,300 m AMSL, model-simulated cloud tops are much more susceptible to entrainment and become more sporadic; this in conjunction with the non-precipitating echo filtering in the MRMS data makes evaluations less meaningful with increasing height.

**Responses to Editor (Page 3)**

9) Check for expressions like "cold temperature". The air is cold. A temperature is not cold, it is low!!

   - Thank you for pointing out this grammatical error. I have scanned through the paper and changed all "cold" temperatures to "low" temperatures. I along with the 4th author also vetted the paper for any other similar logic errors and fixed them.

10) Unfortunately, it happens that scientists read only the abstract and conclusions. To help those readers, you may write the conclusion in a self-contained fashion. I simply mean, that you could again introduce the abbreviations you us, include a link to Table 2. You may also describe in a few more words what GMA, Stage IV precipitation and MRMS is. Is it model or measurement data?

   - Thank you for the advice and suggestions about the abstract and conclusions. While it would be ideal to think a reader would read the whole paper, it probably does not happen most of the time. I have adjusted both the abstract and conclusion to be more self-contained and descriptive as per your suggestion.

[revised manuscript text omitted]

Recent nor'easter studies are scarce given the extensive research efforts of in the 1980s. These historical studies addressed key environmental nor'easter drivers including frontogenesis and baroclinicity (Bosart, 1981; Forbes et al., 1987; Stauffer and Warner, 1987), anticyclones (Uccelini and Kocin, 1987), latent heat release (Uccelini et al., 1987), and moisture transport by the low-level jet (Uccellini and Kocin, 1987; Mailhot and Chouinard, 1989). Despite extensive observational analyses, little ess attention has been given to role of BMPSs in mid-latitude winter cyclones. has been provided to mid-latitude, winter cyclone simulations, especially those focused on BMPSs.

Reisner et al. (1998) ran several several single and double-moment BMPS Mesoscale Model Version 5 winter storm simulations, with multiple BMPS options, of winter storms that impacteding the Colorado Front Range duringfor the Winter Icing and Storms Project. Double moment-based simulations produced more accurate simulations of supercooled water and ice mixing ratios than those originating from single-moment schemes. However, single-moment-based simulations vastly improved when the snow-size distribution intercepts were derived from a diagnostic equation rather than from a fixed value.

Wu and Pretty (2010) investigated how five, six-class BMPSs affected WRF simulations of four polar-low events (two over Japan, two over the Nordic Sea). Their simulations yielded nearly identical storm tracks, but notable cloud top temperature and precipitation errors. Overall, the WRF single-moment BMPS (Hong and Lim, 2006) produced marginally better cloud and precipitation process simulations than compared to those from other BMPSs. For warmer, tropical cyclones, Tao et al. (2011) investigated how four, six-class BMPSs impacted WRF simulations of Hurricane Katrina. They found BMPS choice minimally impacted storm track, yet sea-level pressure (SLP) varied up to 50 hPa.

Shi et al. (2010) evaluated several WRF single-moment BMPSs during a lake-effect snow event. Simulated radar reflectively and cloud top temperature validation revealed that WRF accurately simulated the onset, termination, cloud cover, and band extent of a lake-effect snow event, however snowfall totals at fixed points were less accurate due to interpolation of the mesoscale grid. Inter-BMPS simulation differences were small They found BMPSs produced only minimal simulation differences because lowcold temperatures and weak vertical velocities prevented graupel generation. Reeves and Dawson (2013) investigated WRF sensitivity to eight BMPSs during a December 2009 lake-effect snow event. Simulated Their study found precipitation rates and snowfall coverage were particularly sensitive to BMPSs because vertical velocities exceeded hydrometeor terminal fall speeds in in half of their simulations. Vertical velocity differences were attributed to varying BMPS frozen hydrometeor assumptions concerning snow density values, temperature-dependent snow-intercepts, and graupel generation terms.

We will evaluate WRF nor'easter simulations and their sensitivity to six- and seven-class BMPSs with a focus  on microphysical properties and precipitation.

The remainder of this paper is divided into three sections. Section 2 explains the methodology and analysis methods.

Section 3 shows the results. Finally section 4 describes the conclusions, its implications, and prospects for future research.

**2 Methods**

**2.1 Study design**

We utilized WRF version 3.6.1 (hereafter W361) which solves a set of fully-compressible, non-hydrostatic,

Eulerian equations in terrain-following coordinates (Skamarock et al., 2008). Figure 1 shows the four-domain WRF

grid configuration for this study with a 45-, 15-, 5-, and 1.667-km horizontal grid spacing, respectively.

. Additionally, this configuration includes  61 vertical levels, a 50-hPa (~20 km) model top, two-way domain feedback, and  cumulus parametrization is turned off for Domains 3 and 4 which are convection permitting. Notably, the location of Domain 4 adjusts for each case (Fig. 1).

. Global Forecasting System model operational analysis (GMA) data was used for WRF boundary conditions. The above model configuration (except for the 4[th] domain) and  parameterizations are derived from

Nicholls and Decker (2015).

. Model parameterizations include:

- Longwave radiation: New Goddard Scheme (Chou and Suarez, 1999; Chou and Suarez, 2001)
- Shortwave radiation: New Goddard Scheme (Chou and Suarez, 1999)
- Surface layer: Eta similarity (Monin and Obukhov, 1954; Janjic, 2002)
- Land surface: NOAH (Chen and Dudhia, 2001)
- Boundary layer: Mellor-Yamada-Janjic (Mellor and Yamada 1982; Janjic 2002)
- Cumulus parameterization: Kain-Fritsch (Kain, 2004)

This study investigates the seven nor'easter cases described in Table 1 and shown in Fig. 1. These cases are identical to those in Nicholls and Decker (2015) and represent a small, diverse sample of nor'easter events of varying intensity and seasonal timing.

In Table 1, the Northeast Snowfall Impact Scale (NESIS) value serves as proxy for storm severity (1 = notable,  5 = extreme) and is based upon storm duration,   population impacted, area affected, and snowfall severity (Kocin and Uccellini, 2004). Early and late season storms (Cases 1, 2, and 7) did not have snow and thus lack a NESIS rating.

Five-day, WRF model simulations for this study were initialized 24 hours prior to the first precipitation impacts in the highly populated Mid-Atlantic region and prior to the onset of rapid, coastal cyclogenesis off the North Carolina coastline. This starting point provides sufficient time A 24 hour lead time provides sufficient time for WRF to establish fully develop mesoscale circulations and atmospheric vertical structure (Kleczek et al., 2014), and also to establish key surface baroclinic zones, and sensible and latent heat fluxes (Bosart, 1981; Uccelini and Kocin, 1987; Kuo et al.,

1991; Mote et al., 1997; Kocin and Uccellini, 2004; Yao et al., 2008, Kleczek et al., 2014). We define the first precipitation impact time as the first 0.5 mm (~0.02 inch) precipitation reading from the New Jersey Weather and

Climate Network (D. A. Robinson, pre-print, 2005) associated with a nor'easter event. A smaller threshold was is not used to avoid capturing isolated showers occurring well ahead of the primary precipitation shield.

To investigate BMPS influence upon W361 nor'easter simulations, five BMPS are used (Table 2). These BMPSs include As shown in Table 2, the selected schemes include three, six-class, three-ice, single-moment schemes (Lin

[Lin6; Lin et al., 1983; Rutledge and Hobbs, 1984]), Goddard Cumulus Ensemble [GCE6; Tao et al., 1989; Lang et al., 2007], and WRF single moment [WSM6; Hong and Lim 2006]), a seven-class, four-ice, single-moment Goddard

Cumulus Ensemble scheme (GCE7; Lang et al. 2014), and finally, the a six-class, three-ice, WRF double-moment scheme (WRF double moment, six class (WDM6; Lim and Hong 2010)). In total, 35 model simulations were completed (7 nor'easters x 5 BMPSs).For this study, we ran 35 W361 simulations covering five BMPS and seven nor'easter cases.

**2.2 Evaluation Verification and analysis techniques**

Model evaluation validation and analysis efforts involved comparing WRF model output to focused on comparisons of WRF to GMA, Stage IV precipitation (StIV; Fulton et al. 1998; Y. Lin and K.E. Mitchell, preprints,

2005), and Multi-Radar, Multi-Sensor (MRMS) 3D volume radar reflectivity (Zhang et al. 2016). GMA offers six- hourly, gridded dynamical fields, including water vapor, with global coverage. Stage IV is a six-hourly, 4-km resolution, gridded, combined radar and rain gauge precipitation product covering the United States and is derived from rain gauge and radar data. Finally, MRMS is two minute, 1.3-km resolution, gridded 3D volume radar mosaic product derived from S- and C-band radars covering the United States and Southern Canada (Zhang et al. 2016) and is the . MRMS serves as an the operational successor to the the better known National Mosaic and Multi-Sensor QPE

(NMQ; Zhang et al. 2011) product radar mosaic products. Both Stage IV and MRMS, however are limited by the detection range of their surface-based assets. All cross comparisons between WRF and these evaluation validation data were were conducted at identical grid resolution.

Analysis of WRF model microphysical, precipitation, and simulated radar output was comprised of three main parts: precipitable mixing ratios and domain-averaged mixing ratio profiles, simulated precipitation, and simulated radar reflectivity. Precipitable mixing ratios are calculated for all six microphysical species (vapor, cloud ice, cloud water, snow, rain, and graupel) using the equation for precipitable water:

$$PMR = \frac{1}{\rho g} \int_{P_{top}}^{P_{sfc}} w \, dp \qquad (1)$$

In Eq. (1), *PMR* is the precipitable mixing ratio in mm, $\rho$ is the density of water (1,000 kg m$^{-3}$); $g$ is the gravitational constant (9.8 m s$^{-2}$); $p_{sfc}$ is the surface pressure (Pa), $p_{top}$ is the model top pressure (Pa); $w$ is the mixing ratio (kg kg$^{-1}$); $dp$ is the change in atmospheric pressure between model levels (Pa). We only evaluate water vapor PMR's in this study because all other GMA mixing ratio species are nonexistent  and ground and space validation microphysical data are lacking, especially over the data-poor North Atlantic (Li et al., 2008; Lebsock and Su, 2014). Similarly, mixing ratio profiles will only be inter-compared amongst BMPSs because satellite-derived cloud ice profile products (e.g., CloudSat 2C-ICE; Deng et al. 2013) do not directly overpass Domain 4 during coastal cyclogenesis for any case. WRF-simulated precipitation fields and their distribution were evaluated against StIV and simulation error was quantified via  bias and threat score (critical success index; Wilks, 2011) values. Finally, contoured frequency with altitude diagrams (CFADs) were used to validate WRF-simulated radar reflectivity relative to MRMS similar to  radar validation efforts of Yuter and Houze (1995), Lang et al. (2011) and Lang et al. (2014). A CFAD offers the advantage of preserving frequency distribution information, yet is insensitive to spatio-temporal errors . Additionally, CFAD-based scores were  calculated for each height level and with time  using Eq (2).

$$CS = 1 - \frac{\sum |PDF_m - PDF_o|_h}{200} \tag{2}$$

In (2), *CS* is the CFAD score and $PDF_m$ and $PDF_o$ (%) are the probability density functions (PDF) at constant height from WRF  and MRMS , respectively. The CFAD score ranges between 0 (no PDF overlap) to 1 (identical PDFs).

**3. Results**

**3.1 Hydrometeor species analysis**

Figure 2 displays six classes (water vapor, cloud water, graupel, cloud ice, rain, and snow) of precipitable mixing ratios (mm) from each WRF simulation and GMA  and Fig. 3 shows corresponding simulated radar reflectivity (no MRMS on this date) at 4,000 m above mean sea level (AMSL) from Case 5, Domain 4 at 06 UTC February 2010. We chose this time and height because storm track errors are negligible, the cyclone is centralized within Domain 4, and mixing ratio profiles at this time (Fig. 4) show all hydrometeor species to coincide at 4,000 m AMSL and that snow and graupel mixing ratios approach their maximum values at this height. Figure 5, shows the  seven-case composite mixing ratios derived from hourly data during the residence time each nor'easter case in Domain 4 (24-30 hours). This composite illustrates that mixing ratio profiles  largely preserve their shape, maximum mixing ratio heights, and mixing ratio tendencies (i.e., higher snow mixing ratios in GCE6 and GCE7), but hourly mixing ratio values themselves can vary up to 3.5x's higher (QRAIN; WDM6) at a given height than in the seven case composite (Fig. 5).

Figures 4 and 5 also contain two black dashed lines denoting the 0°C and -40°C heights, which denote the region where super-cooled water may occur. Although both the super-cooled water fraction and these temperature heights vary hourly, the latter demonstrates little to no inter-BMPS variability.

Comparing Figs. 2 and 3  reveals a strong correspondence between radar reflectivity signatures at 4,000 m AMSL and  precipitable hydrometeor species , especially rain, graupel, and snow.  As seen in Fig. 4, all cloud water and rain above 3,500 m AMSL

is super-cooled. Stronger nor'easter-related convection (reflectivity > 35 dBZ) in Fig. 3 best corresponds to precipitable rain and then graupel (Fig. 2) despite the near non-existence of the former at 4,000 m AMSL (Fig. 4).

This apparent discrepancy suggests localized enhancement of rain mixing ratios where stronger vertical velocities near convection likely drive the freezing level higher than Fig. 4 indicates. Within the broader precipitation shield (20-35

dBZ), radar reflectivity patterns best correspond to precipitable snow and then precipitable graupel (Fig. 2) for all

BMPSs except for Lin6 where this trend is reversed. Although Fig. 4 shows all five BMPS loosely agree on amount and height of maximum graupel at 4,000 m AMSL, Lin6 has little to any snow at this level which likely explains the trend reversal.

Inter-BMPS mixing ratio variability  both at this level and throughout the troposphere is due to identifiable trends within the underlying assumptions made by BMPSs and will be explained in more detail below.

All evaluated BMPSs share a common heritage with the Lin scheme  (Note: Lin6 is a modified form of the original Lin scheme). Amongst the BMPSs, only WDM6 explicitly forecasts cloud condensation nuclei, rain, and cloud water number concentrations, the remaining schemes apply derivative equations for these quantities (Hong et al., 2010). Aside from the above, all five BMPS differ primarily in their treatment of frozen hydrometeors which is most evident from the nearly identical (exception: WDM6) rain mixing ratio profiles (Figs. 4 and 5) and precipitable water vapor (Fig. 2) and is a result consistent with Wu and Petty (2010). Comparing WSM6 to WDM6

reveals the second moment has little to no effect on precipitable rain coverage area (Fig. 2) yet, precipitable rain is enhanced (Fig.2) and rain mixing ratios drop sharply near the surface.

[revised manuscript text omitted]

2, 4, and 5. Cloud ice mixing ratios are highest in GCE7 and lowest in Lin6. Wu and Petty (2010) similarly found low cloud ice mixing ratios in Lin6 simulations and ascribe it to dry collection by cloud ice by graupel and its fixed cloud- ice size distribution. Similar to Lin6, GCE6 uses a monodispersed cloud-ice size distribution (20 μm diameter), but assumes vapor growth of cloud ice to snow assuming  water saturation conditions (yet supersaturated with respect ice) leading to higher cloud ice amounts, and  also increased cloud ice to snow conversion rates (Lang et al., 2011; Tao et al., 2016). GCE7 blunts  cloud ice--to--snow conversion rates using a RH correction factor which is dependent upon ice supersaturation which is itself dependent up vertical velocity.

Additionally, GCE7 also includes contact and immersion freezing terms (Lang et al., 2011), makes the cloud ice collection by snow efficiency a function of snow size (Lang et al., 2011; Lang et al., 2014), sets a maximum limit on cloud-ice particle size (Tao et al., 2016), makes ice nuclei concentrations follows the Cooper curve (Cooper, 1986;

Tao et al., 2016), and  allows cloud ice to persist in ice subsaturated conditions (i.e., RH for ice ≥ 70%) (Lang et al,

2011; Lang et al., 2014). Despite the increased cloud ice-to-snow auto conversion rates in GCE7 (Lang et al. 2014;

Tao et al. 2016), precipitable cloud ice amounts nearly doubled relative to GCE6

(See Fig. 2). Similar to GCE7, WSM6  generates larger cloud ice mixing ratios than Lin6, which Wu and Petty (2010) attribute to excess cloud glaciation at temperatures between 0°C

and -20°C and its usage of fixed cloud ice size intercepts. Additionally, both WSM6 and WDM6 include ice sedimentation terms which promote smaller cloud ice amounts (Hong et al., 2008). Despite their varying assumptions, the maximum cloud ice heights for both Case 5 and overall (Figs. 4 and 5) are consistent between BMPSs.

**3.2 Stage IV precipitation analysis**

Excessive precipitation, whether frozen or not, is one of the most potentially crippling impacts of a nor'easter.

Figures 6 and 7 show Domain 3, accumulated precipitation, their difference from St IV, and the associated probability and cumulative distribution functions (PDF and CDF, respectively)  for Cases 5 and 7 based upon the 24-30 hour residence period of a nor'easter within Domain 4.

We focus our attention on Domain 3 for this analysis because most of Domain 4 resides close to or outside the StIV data boundaries.

Cases 5 and 7 are chosen because of their  near-shore tracks (Fig. 1) which affords good StIV data coverage

. Table 3 includes threat score and bias information from  all seven cases and their associated standard deviation statistics. Both threat score and model bias assume the same  10 mm threshold value, which as seen in Figs. 6 and 7 is approximately the 25th percentile of accumulated precipitation on average.

Case 4 threat score and bias values (Table 3) are more than two standard deviations from the composite mean due to its non-coastal storm track (Fig. 1) and thus it is excluded from this analysis

The remaining six cases show WRF to have low-to-moderate forecast skill (Threat score: 0.217 [Lin6] – 0.414 [Lin6]) and to cover too large an area with precipitation values greater than 10 mm

(bias : 1.47 [Lin6, Case 7] – 4.05 [GCE7, Case 3]) relative to StIV. Inter-BMPS threat score and bias differences are an order or magnitude or less than the values from which they are derived.

Consistent with Hong et al. (2010), threat score and bias values from  WSM6 are equal to or improved upon by WDM6 due to its inclusion of a cloud condensation nuclei  feedback. Overall, WDM6 shows marginally better precipitation forecast skill than other BMPSs (lowest threat score in four out of six cases and lowest mean threat score: 0.322), yet Lin6 is the least biased (lowest bias score in four of out of six cases and lowest mean bias: 2.55)generated marginally better simulated precipitation fields and has the lowest threat score in four out of six cases and it also has the lowest model mean (0.322), yet Lin6 was found to be the least bias in four out of six cases and it also has the lowest model mean (2.55).

PDF and CDF plots As illustrated from Figs. 6 and 7 show, all WRF to favor higher precipitation amounts and is consistent with the positive bias scores in Table 3. simulations tended to generate similar coverage to Stage IV, but its precipitation values tended to be smaller than for corresponding grid points in WRF resulting in low to moderate forecast skill and excessively heavy precipitation totals as illustrates in the PDF and CDF diagrams. Previous modelling studies of strong -convection by Ridout et al. (2005) and Dravitzki and McGregor (2011) found that both GFS and the Coupled Ocean/Atmosphere Mesoscale Prediction System (COAMPS) produced too much light precipitation and too much heavy precipitation, which stands in contrast to our results, which show the opposite tendency. Unlike these two studies, our study region lacks nor'easters often track over the data spare North Atlantic, a region with no rain gauge data and is at is near or beyond the operational range limits of S-band radars. These two issues could lead to an under bias in Stage IV precipitation data, especially near the data boundariesedges and suggests that WRF threat scores and biases are , which likely suggests that threat scores and biases are likely closer to observations than as Table 3 indicates shown. Marginal changes in accumulated precipitation PDFs and CDFs (<10 mm) between BMPS simulations and threat scores amongst BMPSs areis consistent with the investigation of simulatedion precipitation during warm-season precipitation events and a quasi-stationary front by (Fritsch and Carbone (, 2004); and Wang and Clark (2010), respectively.

Min (2015)
WDM6 has been reported to reduce light precipitation and increase moderate precipitation, reducing the systematic bias of WSM6 (Hong et al. 2010, Min et al 2015).

using simulated reflectivity products to compare model fields with radar has advantages over radarestimated precipitation fields because there is less uncertainty involved in the calculation of reflectivity from the model than precipitation from radar (Koch et al. 2005; Molthan and Colle 2012)

Among the six hydrometeors, $q_{rain}$, $q_{snow}$, and $q_{graupel}$ are used to calculate the reflectivity.

4-km resolution is needed to account for the complexity of the local topography and to compare directly with radar data

**3.3 MRMS and radar reflectivity analysis**

Figure 8 shows Domain 4, Case 4  radar reflectivity CFADs  constructed over the
24 hour residence time of the nor'easter within Domain 4  (12 UTC 26–27 January 2015). Although not
shown,  the 0°C and -40°C heights are at approximately 2,000 and 8,000 m AMSL, respectively.
We selected Case 4 because its radar data has been reprocessed
with the latest MRMS algorithm, whereas the remaining cases used an older algorithm associated with NMQ and were
still in the process of being updated.
The MRMS CFAD (Fig. 8) shows two, distinct frequency peaks
between 2,300 – 5,000 m and 7,500 m – 11,000 m AMSL, that are not well matched in the models. To investigate
these differences, Figs. 9 and 10 show radar reflectivity at 4,000 and 9,500 m AMSL on 18 UTC 26 January 2015.
Finally, to evaluate model performance against MRMS, Fig. 11 shows a contoured plot of CFAD scores calculated
hourly and at each height level.
CFADs in Fig. 8 depict a distinct bifurcation in the simulated CFADs above and below 6,000 m AMSL relative
to MRMS. Below this level, model-based CFADs generally show a frequency swath that is broader than MRMS and
overly favors the occurrence of stronger reflectivity values (exceptions: GCE7 and Lin6). Above this level, CFADs
display a reflectivity frequency swath that is generally too narrow (exception: GCE6) and favors weaker reflectivity
values relative to MRMS. Below the 0°C height level (2,000 m AMSL), all models over extend the reflectivity
frequency range (5% frequency: Models = -15 – 32 dBZ; MRMS = -1 – 27 dBZ), yet only Lin6 and especially GCE7
correctly capture the maximum frequency core between 5 and 15 dBZ. Other schemes produce this core, but it is offset
by 10 dBZ or more toward higher reflectivity values. Case 4 mixing ratio profiles (not shown) depict similar
relationships amongst the hydrometer species as shown for Case 5 (Fig. 4), albeit with a comparably lower freezing
level. Below the freezing level, both GCE7 and Lin6 have lower graupel mixing ratios values than other schemes.
Given the earlier correspondence (Section 3.1) between graupel and stronger reflectivity values this does suggest a
probable explanation for the better results of Lin6 and GCE7. At higher altitudes (2,000 – 6,000 m), both Lin6 and
GCE7 maintain the best representation of the maximum frequency core with radar reflectivity values that are indeed
lower than other schemes and closer to MRMS (Fig. 9 at 4,000 m AMSL). Above 6,000 m AMSL, WRF CFADs shift
toward very low reflectivity values (< 0 dBZ) due to increased entrainment near the simulated cloud top. This shift if
particularly pronounced in GCE7 where the combination of its new temperature and mixing ratio dependent
aggregation rates and snow map produce smaller hydrometeors are lower temperatures and larger hydrometeors at
higher temperatures. Near the top of the troposphere (> 7,500 m AMSL), CFADs values are based solely upon
increasingly isolated reflectivity values (Fig. 10 at 9,500 m AMSL) which leads to the notable discrepancies in CFAD
structure between the models and MRMS.

Consistent with the above discussion, CFAD scores with height and time (Fig. 11) show Lin6 to qualitatively perform best overall, however, GCE7 simulations below 5,000 m AMSL typically attained even higher CFAD scores.

Other BMPSs (as shown in Fig. 8) typically favor unrealistically high distribution of reflectivity values and also exhibit lower CFAD scores in the melting layer likely due to  higher graupel mixing ratios. Further aloft, aggregation of hydrometeors toward smaller sizes and entrainment likely cut off cloud tops in GCE7 more so than in other schemes and results in its lower CFAD scores above 6,000 m

AMSL. The other six cases produce similar tendencies in their CFAD and CFAD scores as noted above for Case 4, except cloud heights become higher and CFADs become wider with the introduction of stronger convection in early and late season events.

**4 Conclusions**

The role and impact of five bulk microphysics schemes (BMPS; Table 2) upon seven, Weather Research and

[revised manuscript text omitted]

Hong, S -Y., and Lim, J. -O. .J.: The WRF single-moment 6-class microphysics scheme (WSM6), J. Korean Meteor.
Soc., 42, 129-151, 2006.

Hong, S. -Y., Lim, K. -S. S., Lee, Y. -H., Ha, J. -C., Kim, H. -W., Ham, S. -J., and Dudhia, J.: Evaluation of the
WRF double-moment 6-class microphysics scheme for precipitating convection, Adv. Meteor., 2010,
doi:10.1155/2010/707253, 2010.

Hou, A. Y., Kakar, R. K., Neeck, S., Azarbarzin, A. A., Kummerow, C. D., Kojima, M., Oki, R., Nakamura, K., and
Iguchi, T.: The Global Precipitation Measurement Mission, Bull. Amer. Meteor. Soc., 95, 701–722, 2014.

Jacobs, N. A., Lackmann, G. M., and Raman, S.: The combined effects of Gulf Stream-induced baroclinicity and
upper-level vorticity on U.S. East Coast extratropical cyclogenesis, Mon. Wea. Rev., 133, 2494–2501, 2005.

Janjic, Z. I.: Nonsingular implementation of the Mellor–Yamada level 2.5 scheme in the NCEP meso model, NCEP
Office Note 437, 61 pp., 2002.

Justice, C. O. et al. (1998), The Moderate Resolution Imaging Spectroradiometer (MODIS): land remote sensing for
global change research, IEEE Transactions on Geoscience and Remote Sensing, 36, 1228–1249, 1998.

Kain, J. S.: The Kain–Fritsch Convective Parameterization: An Update, J. Appl. Meteor., 43, 170–181, 2004.

Kessler, E.: On the distribution and continuity of water substance in atmospheric circulation, Meteor. Monogr., 32,
Amer. Meteor. Soc., 84 pp, 1969.

Kleczek, M. A., G.-J. Steenveld, and A. A. M. Holtslag:  Evaluation of the Weather Research and Forecasting
Mesoscale Model for GABLS3: Impact of boundary-layer schemes, boundary conditions and spin-up,
Boundary-Layer Meteorol, 152, 213–243, 2014.

Kocin, P. J. and Uccellini, L. W.: Northeast snowstorms. Vols. 1 and 2, Meteor. Monogr., No. 54., Amer. Met. Soc.,
818 pp., 2004.

Kuo, Y. H., Low-Nam, S., and Reed, R. J.: Effects of surface energy fluxes during the early development and rapid
intensification stages of seven explosive cyclones in the Western Atlantic. Mon. Wea. Rev., 119, 457–476, 1991.

Lang, S., Tao, W. -K., Cifelli, R., Olson, W., Halverson, J., Rutledge, S., and Simpson, J.: Improving simulations of
convective system from TRMM LBA: Easterly and westerly regimes, J. Atmos. Sci., 64, 1141–1164, 2007.

Lang, S. E., Tao, W. -K., Zeng, X., and Li, Y.: Reducing the biases in simulated radar reflectivities from a bulk
microphysics scheme: Tropical convective systems, J. Atmos. Sci., 68, 2306–2320, 2011.

Lang, S. E., Tao, W. -K., Chern, J. -D., Wu, D., and Li, X.: Benefits of a fourth ice class in the simulated radar
reflectivities of convective systems using a bulk microphysics scheme, J. Atmos. Sci., 71, 3583–3612,
doi:10.1175/JAS-D-13-0330.1, 2014.

Lebsock, M., and Su, H: Application of active spaceborne remote sensing for understanding biases between passive
cloud water path retrievals, J. Geophys. Res. Atmos., 119, 8962–8979, doi:10.1002/2014JD021568, 2014.

Li, J. -L. F., Waliser, D., Woods, C., Teixeira, J., Bacmeister, J., Chern, J. -D.,, Shen, B. -W., Tompkins, A., Tao,
W. -K.,, and Kohler, M.: Comparisons of satellites liquid water estimates to ECMWF and GMAO analyses,
20th century IPCC AR4 climate simulations, and GCM simulations, Geophys. Res. Lett., 35, L19710,
doi:10.1029/2008GL035427, 2008.

Lim, K.-S. and Hong, S. -Y.: Development of an effective double-moment cloud microphysics scheme with
prognostic cloud condensation nuclei (CCN) for weather and climate models, Mon. Wea. Rev., 138, 1587–
1612, 2010.

Lin, Y. -L., Farley, R. D., and Orville, H. D.: Bulk parameterization of the snow field in a cloud model, J. Climate
Appl. Meteor., 22, 1065–1092, 1983.

Mailhot, J. and Chouinard, C.: Numerical forecasts of explosive winter storms: Sensitivity experiments with a meso-
scale model, Mon Wea. Rev., 117, 1311–1343, 1989.

Marzban C., and Sandgathe, S.: Cluster analysis for verification of precipitation fields, Wea. Forecasting, 21, 824–
838, 2006.

Mellor, G. L., and Yamada, T.: Development of a turbulence closure model for geophysical fluid problems, Rev.
Geophys. Space Phys., 20, 851–875, 1982.

Min, K.-H., S. Choo, D. Lee and G. Lee: Evaluation of WRF cloud microphysics schemes using radar observations.
Weather and Forecasting, 30, 10.1175/WAF-D-14-00095.1, 2015.

Monin, A. S., and Obukhov, A. M.: Basic laws of turbulent mixing in the surface layer of the atmosphere. Tr. Akad.
Nauk SSSR Geophiz. Inst., 24, 163–187, 1954.

Morath, E. (2016), Will a blizzard freeze U.S. economic growth for the third straight year, Wall Street Journal, 20
Jan. 2016.

Morrison, H., Thompson, G., and Tatarskii, V.: Impact of cloud microphysics on the development of trailing stratiform
precipitation in a simulated squall line: Comparison of one- and two-moment schemes, Mon. Wea. Rev., 137,
991–1007, 2009.

Mote, T. L., Gamble, D. W., Underwood, S. J., Bentley, M. L.: Synoptic-scale features common to heavy snowstorms
in the Southeast United States, Wea. Forecasting, 12, 5–23, 1997.

Nicholls, S. D. and Decker, S. G.: Impact of coupling an ocean model to WRF nor'easter simulations, Mon. Wea.
Rev., 143, 4997–5016, 2015.

Reeves, H. D. and Dawson II, D. T.: The dependence of QPF on the choice of microphysical parameterization for
lake-effect snowstorms, J. Appl. Meteor. Climatol., 52, 363–377, 2013.

Reisner, J. R., Rasmussen, R. M., and Bruintjes, R .T.: Explicit forecasting of supercooled liquid water in winter
storms using the MM5 mesoscale model. Quar. J. Roy. Met. Soc., 124, 1071-1107, 1998.

Ridout, J. A., Y. Jin, and Liou, C. -S.: A cloud-base quasi-balance constraint for parameterized convection:
Application to the Kain–Fritsch cumulus scheme, Mon. Wea. Rev., 133, 3315–3334, 2005.

Rutledge, S. A., and Hobbs, P. V.: The mesoscale and microscale structure and organization of clouds and precipitation
in mid-latitude cyclones. XII: A diagnostic modeling study of precipitation development in narrow cloud-frontal
rainbands. J. Atmos. Sci., 20, 2949–2972, 1984.

Shi, J. J. et al.: WRF simulations of the 20-22 January 2007 snow events of Eastern Canada: Comparison with in situ
and satellite observations, J. Appl. Meteor. Climatol., 49, 2246–2266, 2010.

Skamarock, W.C., Klemp, J. P., Dudhia, J., Gill, D. O., Barker, D. M., Duda, M. G., Huang, X. -Y., Wang, W., and
Powers, J. G.: A description of the advanced research WRF version 3, NCAR Tech. Note NCAR/TN–475+STR,
125 pp., 2008.

Smith, A. B., and Katz, R. W.: US billion-dollar weather and climate disasters: Data sources, trends, accuracy and
biases, Natural Hazards, 67, 387–410, 2013.

Stark, D.: Field observations and modeling of the microphysics within winter storms over Long Island, NY. M.S.
thesis, School of Marine and Atmospheric Sciences, Stony Brook University, 132 pp., 2012.

Stauffer, D. R., and Warner, T. T.: A numerical study of Appalachian cold-air damming and coastal frontogenesis,
Mon. Wea. Rev., 115, 799–821, 1987.

Stephens, G. L., et al.: CloudSat mission: Performance and early science after the first year of operation, J. Geophys.
Res., 113, D00A18, doi:10.1029/2008JD009982, 2008.

Stith, J. L., Dye, J. E., Bansemer, A., Heymsfield, A. J., Grainger, C. A., Petersen, W. A, and Clfelli, R.:
Microphysical observations of tropical clouds, J. Appl. Meteor., 41, 97–117, 2002.

Tao, W. -K., Simpson, J. and McCumber, M.: An ice-water saturation adjustment, Mon. Wea. Rev., 117, 231–235,
1989.

Tao, W. -K., Shi, J. J., Chen, S. S., Lang, S., Lin, P. -L., Hong, S. -Y., Peters-Lidard, C., and Hou, A.: 
[revised manuscript text omitted]